# An injectable liposome-anchored teriparatide incorporated gallic acid-grafted gelatin hydrogel for osteoarthritis treatment

Guoqing Li [1,2,5], Su Liu[1,2,5], Yixiao Chen[1,2,5], Jin Zhao[1,2], Huihui Xu[1,2], Jian Weng[1,2], Fei Yu[1,2], Ao Xiong[1,2], Anjaneyulu Udduttula[3], Deli Wang[1,2], Peng Liu [1,2] ✉, Yingqi Chen [1,2] ✉ & Hui Zeng [1,2,4] ✉

Intra-articular injection of therapeutics is an effective strategy for treating osteoarthritis (OA), but it is hindered by rapid drug diffusion, thereby necessitating high-frequency injections. Hence, the development of a biofunctional hydrogel for improved delivery is required. In this study, we introduce a liposome-anchored teriparatide (PTH (1–34)) incorporated into a gallic acid-grafted gelatin injectable hydrogel (GLP hydrogel). We show that the GLP hydrogel can form in situ and without affecting knee motion after intra-articular injection in mice. We demonstrate controlled, sustained release of PTH (1–34) from the GLP hydrogel. We find that the GLP hydrogel promotes ATDC5 cell proliferation and protects the IL-1β-induced ATDC5 cells from further OA progression by regulating the PI3K/AKT signaling pathway. Further, we show that intra-articular injection of hydrogels into an OA-induced mouse model promotes glycosaminoglycans synthesis and protects the cartilage from degradation, supporting the potential of this biomaterial for OA treatment.

Osteoarthritis (OA) is a degenerative and debilitating joint disease characterized by progressive cartilage degeneration, subchondral bone (SCB) remodeling, synovitis, and osteophyte formation[1,2]. The etiologies of OA are complex and involve a variety of factors, such as genetic predisposition, acute injury, and chronic inflammation. The incidence of OA increased with age, and the main manifestations are persistent pain and limited mobility, which seriously impair the patient's physical function and quality of life[3]. In clinical practice, the commonly used treatment methods include the usage of analgesics and anti-inflammatory drugs, such as non-steroidal anti-inflammatory drugs, cyclooxygenase-2 inhibitors, and acetaminophen, paracetamol, or even opioids, which alleviate OA symptoms in the shortterm, but are unable to reverse the progression of OA[4,5]. At the same time, the traditional system of administration, such as oral and intravenous injections, suffers from a low utilization rate and a potential risk of systemic side effects. However, as a promising method for OA treatment, topical administration, such as intra-articular injection, can minimize the aforementioned problems[6]. In topical administration, the drug carrier form determines the drug release profile and the frequency of intra-articular injection, and is thus a critical factor. In the majority of existing studies, the carrier for bioactive components in OA treatment has been balanced salt solutions or nanoparticles, which largely failed in lab and clinical evaluations due to the rapid diffusion of the injected carrier and fast release of bioactive components, leading to frequent injections[7–9]. Notably, frequent injection severely reduces the effect of the drug and increases the incidence of adverse events

[1]Department of Bone & Joint Surgery, Peking University Shenzhen Hospital, Shenzhen 518036, PR China. [2]National & Local Joint Engineering Research Center of Orthopaedic Biomaterials, Peking University Shenzhen Hospital, Shenzhen 518036, PR China. [3]Centre for Biomaterials, Cellular and Molecular Theranostics (CBCMT), Vellore Institute of Technology (VIT), Vellore, Tamil Nadu 632014, India. [4]Present address: Department of Bone & Joint Surgery, Peking University Shenzhen Hospital, 1120 Lianhua Road, Futian District, Shenzhen, Guangdong Province, PR China. [5]These authors contributed equally: Guoqing Li, Su Liu, Yixiao Chen. ✉e-mail: liupeng_polymer@126.com; yqchen0203@foxmail.com; zenghui_36@163.com

such as bleeding, joint infection, or even systemic side effects[10,11]. In overcoming the described issues, injectable biomaterials with good biocompatibility, biodegradability, drug loading, and controlled release properties have shown considerable promise in the treatment of OA, being minimally invasive and able to guarantee the long-term local release of drugs[12].

The practical OA situation should be considered in the design and fabrication of injectable biomaterials for OA treatment. In a normal joint, the chondrocytes are in an amorphous extracellular matrix (ECM) that is filled with collagen and proteoglycans, and such an environment is essential for the normal function of chondrocytes, namely the proliferation and synthesis of collagen type II (COLIIA1) and aggrecan (ACAN). However, as an abnormal situation, OA does not just involve the wear-and-tear of cartilage, and is increasingly regarded as a whole joint disorder involving the remodeling of the SCB and synovium[13]. The initiation of OA could potentially be induced by adverse mechanical loading, wherein the ingredients result in a proper response by producing catabolic and inflammatory mediators produced by chondrocytes, which could potentially lead to synovitis or even aggravate the OA progression[14,15]. The normal environment of chondrocytes would be disrupted in OA with the overexpression of catabolic enzymes, which causes degradation of the ECM, such as a decrease in glycosaminoglycans (GAG)[16] and degradation of normal cartilage[17]. The degradation of ECM will increase the frictional wear of the articular cartilage, causing pain and even leading to disability for OA patients. As such, strategies for maintaining a normal intra-articular environment, such as inhibiting aberrant hypertrophic maturation and inducing matrix production are generally regarded as promising therapeutic modalities for OA treatment, which may mobilize the cells in the joint cavity for OA treatment. From such a perspective, aside from the aforementioned properties, the ideal injectable biomaterials should possess other attractive properties, such as the ability to promote synthesis of the ECM, protect normal function, and promote the proliferation of chondrocytes, which would be an alternative strategy for OA treatment with great potential.

Among the injectable biomaterials, the hydrogels made from natural polysaccharides, such as chitosan, gelatin, chondroitin sulfate, alginate, and others, have shown potential in biomedical applications[18]. Such potential can be attributed to the natural polysaccharides usually have similar chemical components or structures to natural ECM, and thus, such polymers naturally possess good biosafety, biocompatibility, biodegradability, and even an anti-inflammatory property, which are attractive properties in the treatment of OA. Among the aforementioned natural polysaccharides, gelatin is a type of denatured collagen, and many studies have confirmed its merits, such as its affinity for chondrocyte cells, and the biocompatibility of gelatin-based hydrogels in the treatment of OA[19]. However, the focus of recent research has been on the improvement of mechanical properties and chondrogenic-promotive properties of gelation-based injectable hydrogels, namely increasing the cross-linking density by introducing a covalent bond (usually containing non-biodegradable or even toxic components) into gelatin hydrogels or loading with drugs (for example, oxidized dextran) for cartilage defect repair[20]. Despite such research efforts, to the present knowledge, there is a scarcity of studies in which the development of gelatin-based injectable hydrogels (without any toxic and non-degradable components) has been achieved using a mild and simple fabrication method that can maintain the normal function of chondrocytes and promote the synthesis of the ECM in the joint cavity for OA treatment.

Teriparatide (PTH (1–34)), an active recombinant human peptide sequence of the parathyroid hormone, an Food and Drug Administration-approved drug for osteoporosis treatment[21], is a parathyroid hormone analog that stimulates chondrocyte proliferation and differentiation in addition to cartilage regeneration[22]. PTH (1–34) inhibits the terminal differentiation of articular chondrocytes and, in turn, suppresses the progression of OA[23], exhibiting a protective effect for both articular cartilage and SCB in OA animals[24]. Additionally, intra-articular injection of PTH (1–34) has been found to improve knee function in ageing-related OA[25]. However, the water solubility of PTH (1–34) is high, which results in rapid diffusion when dissolved in a balanced salt solution for intra-articular injection. Thus, a system that can facilitate loading and control of the release of water-soluble PTH (1–34) is a promising strategy for OA treatment.

Owing to the aforementioned factors, in the present study, PTH (1–34) was encapsulated in the core of liposomes by means of a film dispersion method, and then the liposomes loaded with PTH (1–34) (Lipo@PTH (1–34)) were incorporated into the gallic acid-grafted gelatin (GGA) solution, followed by the formation of the injectable hydrogel by means of the transglutaminase (TG) enzymatic cross-linking method. To the best of our knowledge, the present study is one of the cutting-edge attempt in which PTH (1–34) was incorporated into such a fully-biodegradable, biocompatible natural polysaccharide-based hydrogel system to control release, with the aim of alleviating OA progression and protecting chondrocytes degradation. Compared with reported particle-loaded hydrogel system[26], the present hydrogel is non-toxic and the major components are natural components, and the fabrication process is mild and controllable (no need for harsh conditions), having great potential in clinic usage and industry. In the present hydrogel system, gallic acid, a small natural antioxidant small phenolic molecule, was grafted onto gelatin to obtain the GGA through the crosslinking of amino groups in gelatin and carboxyl groups in gallic acid. As a significant gelatin derivative, GGA has attracted considerable attention for biomedical applications. In addition to the attractive properties of gelatin, GGA possesses potential reactive oxygen species (ROS)-scavenging and anti-inflammatory properties[27], since the phenolic hydroxyl group, especially the non-oxidized ones in GGA might consume the ROS in the damaged tissue area. As such, the focus of recent studies on GGA hydrogel has been on the ROS-scavenging-related tissue regeneration and anti-inflammatory properties, including the promotion of diabetic wound healing[28], neural repair in traumatic brain injury[29], and other tissue regeneration. OA is a disease that is primarily caused by degeneration and inflammation, and thus, the use of GGA hydrogel as a matrix material for drug loading and release is desirable.

Here, we develop a liposome loading PTH (1–34) incorporated GGA-based injectable hydrogel (GGA@Lipo@PTH (1–34), GLP) with TG enzymic-linking method, and explore the potential applications in OA treatment. A schematic illustration of the fabrication of the injectable hydrogel system and the main research idea of this study is presented in Fig. 1. Based on our results we (i) show that the GLP hydrogel can be formed via intra-articular injection in the mice, and the knee motion would not be affected, (ii) evidence that PTH (1–34) is released in a sustained manner from the GLP hydrogel when intra-articular injected in mice, (iii) verify that the GLP hydrogel promote ATDC5 cells proliferation and protect the IL-1β-induced ATDC5 cells from further progression potentially through PI3K/AKT signaling pathway, and (iv) confirm that the GLP hydrogel can protect cartilage from degradation, alleviate OA progression, and promote the sysnthesis of GAG when intra-articular injection in DMM mice.

## Results
### ¹H NMR for gallic acid-grafted gelatin

The synthesis process of GGA and the ¹H NMR results are presented in Supplementary Fig. 1. In the ¹H NMR spectra, a new characteristic peak was detected at 7.0 part per million (PPM) which corresponded to the gallic acid in the GGA component, but such a peak was not found in gelatin. From the ¹H NMR results, the successful grafting of gallic acid onto gelatin was verified, which was in consistent with reported study[27].

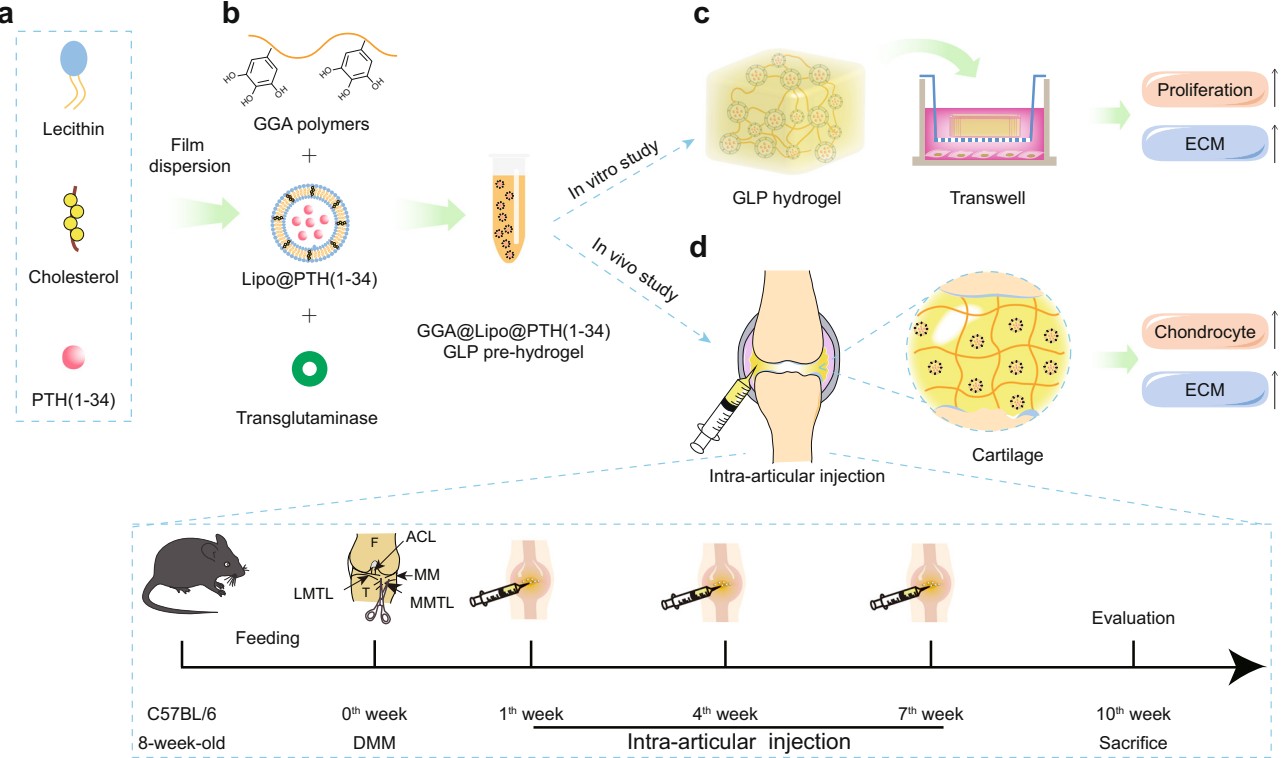

**Fig. 1 | Schematic illustration for the present study. a** The preparation of liposomes loaded with PTH (1–34) (Lipo@PTH(1–34)) by means of the film dispersion method. **b** The preparation of GGA@Lipo@PTH (1–34) (GLP) hydrogel. **c** In vitro study for the GLP hydrogel including physico-chemical properties characterization and in vitro ATDC5 cells culture study. **d** In vivo intra-articular injection of the GLP hydrogel and the treatment protocol for the in vivo study. ECM extracellular matrix, DMM destabilization of the medial meniscus.

## Characterization of Lipo@PTH (1–34) and GLP hydrogel

The loading efficiency of PTH (1–34) in liposomes was characterized with the High Performance Liquid Chromatography-Mass Spectrometry (HPLC-MS, Thermo Exploris 480, US) method. The loading efficiency of PTH (1–34) in liposomes was optimized by inputting different amounts of PTH (1–34) in the same amount of liposome (Supplementary Fig. 2). When the input amount of PTH (1–34) was 1.0 mg, the loading efficiency of PTH (1–34) in liposomes was ca. 66.5%, and with the input amount of PTH (1–34) increased to 1.5 mg, the loading efficiency of PTH (1–34) was increased to ca. 72%, but loading efficiency was not further increased when the input amount increased to 2.0 mg. Therefore, the 1.5 mg PTH (1–34) input amount was utilized for the further investigations.

The characterization results for Lipo@PTH (1–34) and GLP hydrogel were presented in Fig. 2. The transmission electron microscope (TEM) image of the Lipo@PTH (1–34) nanoparticles in Fig. 2a showed that the size of vesicle-like particles was ca. 60–100 nm; the particle size results in Fig. 2b indicated that the average particle size for Liposome and Lipo@PTH (1–34) was ca. 200 nm, larger than the TEM observation, which could be attributed to the particles being hydrated during the particle size detection. In Fig. 2c, the zeta potential for Lipo@PTH (1–34) was ca. 12 mV, larger than −13 mV for liposome particles.

The GLP hydrogel characterization results are shown in Fig. 2d–i. The macroscopic images for the gelation of hydrogels in Fig. 2d showed that after the inclusion of 0.5 wt% TG into GGA and GLP prehydrogel solutions, the hydrogels would form in ca. 5 min at 37 °C. The time sweep rheological results in Fig. 2e revealed that at the initial stage, the elastic moduli G′ for GGA and GLP hydrogels were lower than the viscous moduli G″. However, the elastic moduli G′ for the hydrogels were significantly increased over time, while the viscous moduli G″ for the hydrogels increased slowly. After ca. 160 seconds, the elastic

moduli for GGA and GLP hydrogels were larger than the viscous moduli, indicating the formation of the hydrogels (sol-gel transformation). After 300 seconds, the elastic moduli for hydrogels were 10 times larger than the viscous moduli, confirming the formation of stable hydrogels. The frequency sweep results in Fig. 2e showed that the elastic moduli G′ for GGA and GLP hydrogels were 10 times larger than those of the viscous moduli G″, indicating the formation of stable hydrogels.

The macroscopic images of the fabricated hydrogels after being allowed to swell in deionized (DI) water for 5 days are presented in Fig. 2f. The GGA and GLP hydrogels exhibited no significant difference in volume for 5 days in PBS or DI water immersion. The surface of the scanning electron microscope (SEM) images (in Fig. 2g) showed that the lyophilized hydrogel presented a porous structure, and the pore size ranged from 40 to 100 μm. The chemical components of the hydrogels were investigated by means of Fourier transformed infrared (FTIR) and X-ray photoelectron spectroscope (XPS) assessments, and the results are presented in Fig. 2h, i. In the FTIR spectrum, the characteristic peaks for C-O-C at 1030 cm⁻¹, C-O at 1230 cm⁻¹, C-N at 1440 cm⁻¹, -C=O- at 1630 cm⁻¹, C-H at 2930 cm⁻¹, and -CO-NH- at 3300 cm⁻¹ bands were detected in the two hydrogels, which were obtained from the GGA hydrogel. However, there were no obvious characteristic peaks for Lipo@PTH (1–34) in the GLP hydrogel, which could be attributed to the low amount of Lipo@PTH (1–34) nanoparticles as well as the low sensitivity of the FTIR test. The XPS spectra in Fig. 2i showed that the characteristic peak of P 2p at 133.0 eV from the liposomes was detected in the GLP hydrogel, but such a peak was not detected in the GGA hydrogel, confirming the existence of liposomes in the GLP hydrogel.

Besides, the different concentrations of Lipo@PTH (1–34) were incorporated into hydrogels to investigate the PTH (1–34) release profile, and the results were presented in Supplementary Fig. 3. As

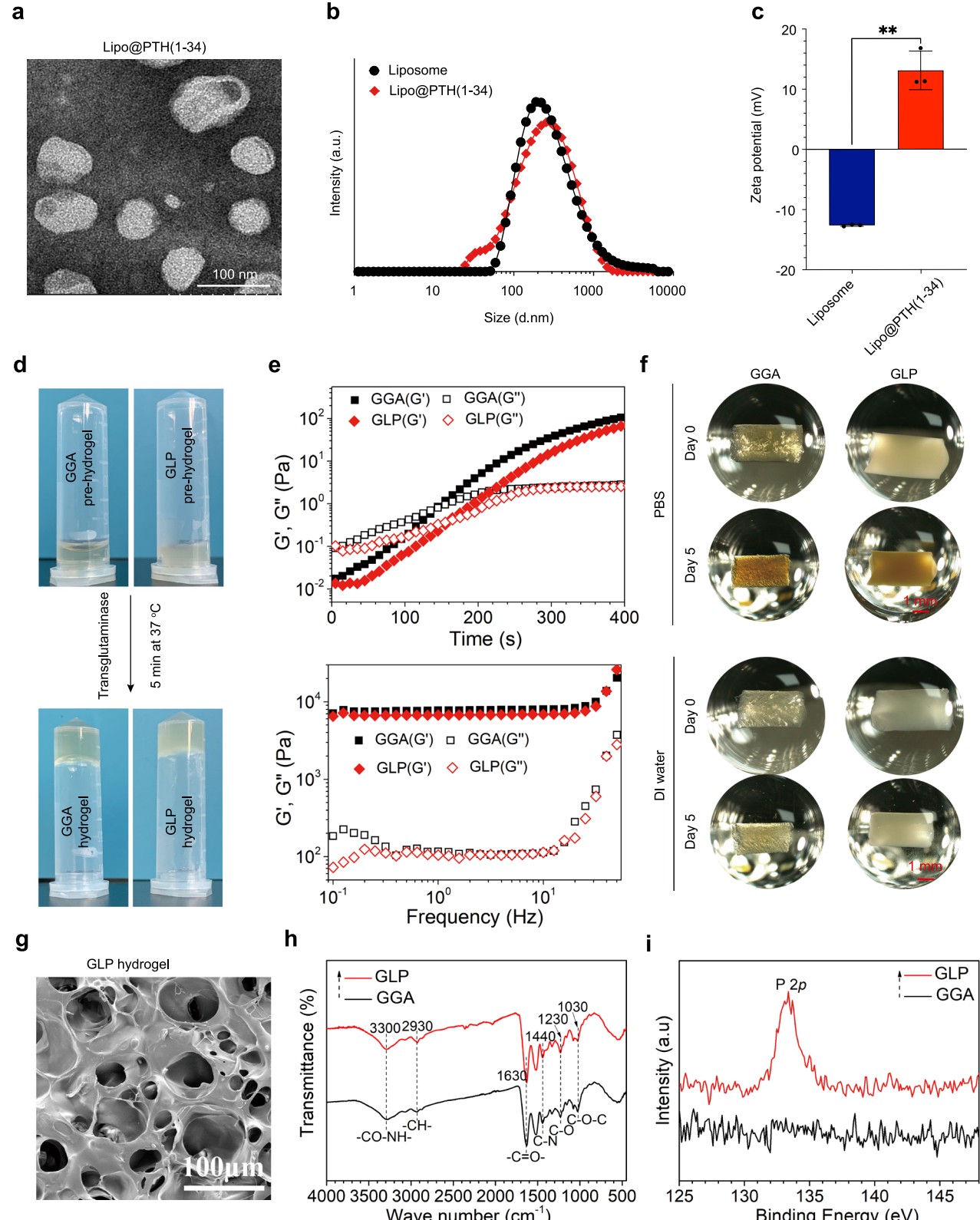

**Fig. 2 | Characterization for Lipo@PTH (1–34) and the GLP hydrogel versus GGA hydrogel. a** TEM image for the Lipo@PTH (1–34). **b** The particle size for the liposomes and Lipo@PTH (1–34). **c** The Zeta potential for the liposomes and Lipo@PTH (1–34) particles. Data are represented as means ± SD of three replicate experiments. **d** The gelation of GGA and GLP hydrogels using the transglutaminase enzymatic method. **e** Rheological properties of hydrogel, time sweeping of elastic moduli G' and viscous moduli G" for samples under the strain of 1% and temperature of 37 °C (up), frequency sweeping of elastic moduli G' and viscous moduli G" for samples under the strain of 1% and temperature of 37 °C (below). **f** Macroscopic images for the hydrogels before and after swollen in PBS and DI water for 5 d. **g** SEM image for the lyophilized GLP hydrogel. **h, i** FTIR and XPS spectra for the GGA and GLP hydrogels. (n = 3 independent experiments). A.u arbitrary units, PBS phosphate-buffered saline, GGA gallic acid-grafted gelatin, GLP GGA@Lipo@PTH (1–34), DI deionized.

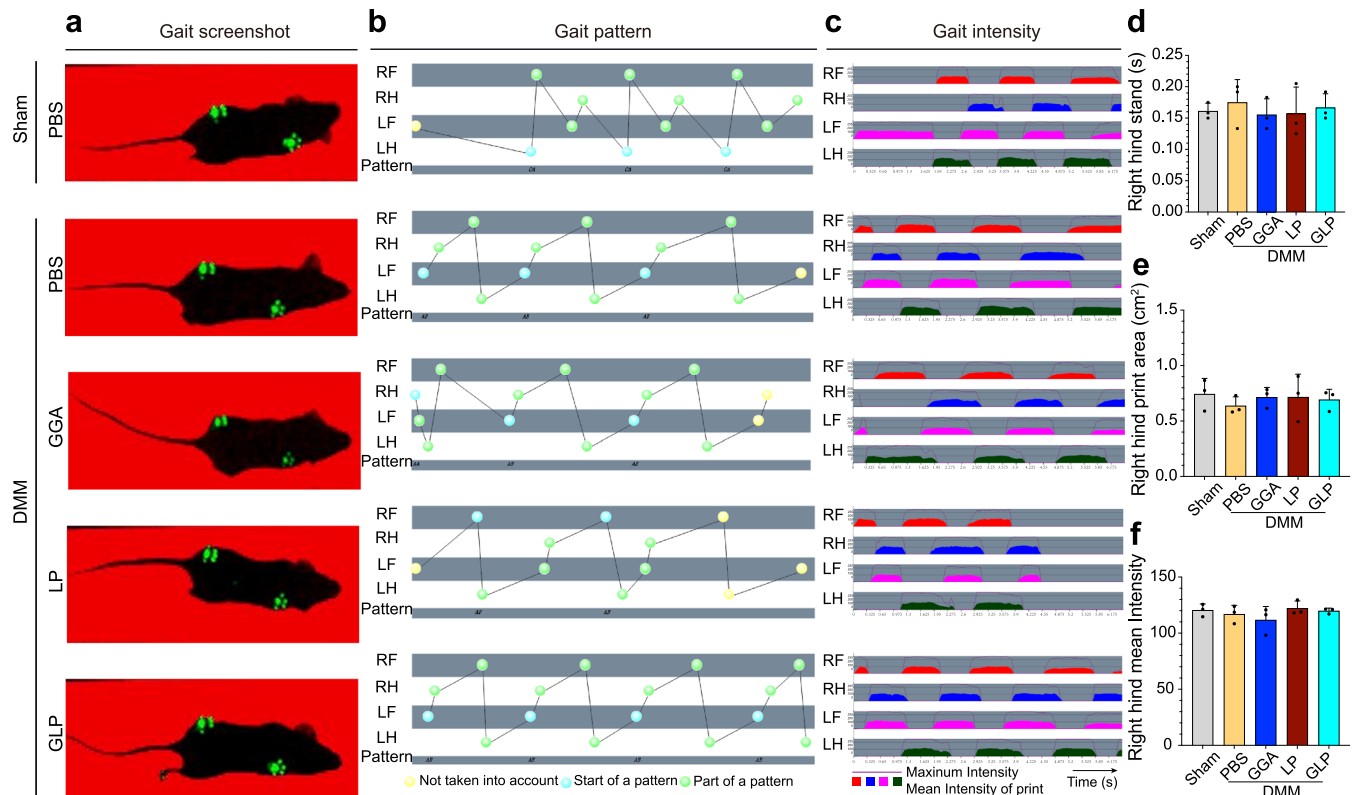

**Fig. 3 | Gait analysis for the mice after intra-articular injection with hydrogels.** **a** Representative gait screen shot. **b** Representative gait pattern. **c** Representative gait intensity. **d** Quantification of right hind stands. **e** Quantification of right hind print area. **f** Quantification of right hind mean intensity. Data are represented as means ± SD of at least three replicate experiments. DMM destabilization of the medial meniscus, PBS phosphate-buffered saline, GGA gallic acid-grafted gelatin, LP Lipo@PTH(1–34), GLP GGA@Lipo@PTH (1–34), RF right forearm, LF left forearm, RH right hind, LH left hind.

shown in In Vivo Imaging System (IVIS) images of hydrogels (Supplementary Fig. 3a), the higher amount of Lipo@PTH (1–34) incorporated hydrogels (1 wt% and 0.5 wt%) showed higher radiant intensity in comparison with the lower amount of Lipo@PTH (1–34) incorporated hydrogels, and with the increase of immersion time the radiant intensity was gradually decreased. The Lipo@PTH (1–34) (1 wt%) incorporated hydrogel showed the largest radiant intensity in all these detected time points. The statistical analysis result in Supplementary Fig. 3b confirmed the IVIS observation. Meanwhile, the accumalted PTH (1–34) release in immersed phosphate-buffered saline (PBS) solutions (Supplementary Fig. 3c) showed the same tendency as IVIS images in hydrogels.

The in vitro degradation of hydrogels was performed by immersing the samples in PBS at 37 °C in a shaking state for up to 28 days, and the results were shown in Supplementary Fig. 4. The mass of GGA and GLP hydrogels gradually decreased with the progress of immersion time.

### Mechanical properties of hydrogels

The mechanical properties results were presented in Supplementary Fig. 5. As shown in Supplementary Fig. 5b, the rectangular hydrogels would be recovered to their initial state after 360° rotation test. From the stretching test results (Supplementary Fig. 5c, d), these two hydrogels would not be broken until stretching to four times their original length, and from the tensile stress vs. strain curves, the tensile strength for GGA and GLP hydrogels was ca. 20.0 kPa and ca. 17.5 kPa, respectively. The compressive stress vs. strain curves in Supplementary Fig. 5g showed that these two hydrogels would be broken after being compressed down to ca. 75% of their initial height, and the compressive strength for these two hydrogels was ca. 250 kPa.

### In situ colloidal formation of hydrogel via intra-articular injection in the joint cavity

The anatomy method was adopted for investigating the colloidal formation of hydrogels in the joint cavity, and the results are presented in Supplementary Fig. 6. Before intra-articular injection, the pre-hydrogel solution or PBS solution (as a control) was mixed with carbon nano-particles suspension injection. As shown, an observation could be made that after intra-articular injection, the stained area was aggregated in the joint cavity for the hydrogel group, but diffused for the PBS control group with the expression from the joint cavity. Further, a gel-like substance was present in the joint cavity for the hydrogel group, but no solid-like substance was observed in the PBS group. Such results demonstrated the colloidal formation of the hydrogels after intra-articular injection.

### Gait analysis for the mice after intra-articular injection with hydrogels

The gait analysis for mice after intra-articular injection with hydrogels was performed to evaluate the knee motion, and the results are displayed in Fig. 3. As shown, the gait index, including gait screenshots, gait patterns, and gait intensities, presented no significant difference among the samples. At the same time, the right hind that was intra-articularly injected with hydrogels was analyzed independently, and the parameters indicating the right hind functions, including stand, print area, and mean intensity, were obtained. The aforementioned parameters showed no significant differences among the samples. The gait analysis results indicated that the intra-articular injection of hydrogels would not affect the knee motion. In addition, the knee range of motion (ROM) for the mice after intra-articular injection with samples was also investigated, and the results are shown in Supplementary Fig. 7. From the images, there were no differences in the ROM

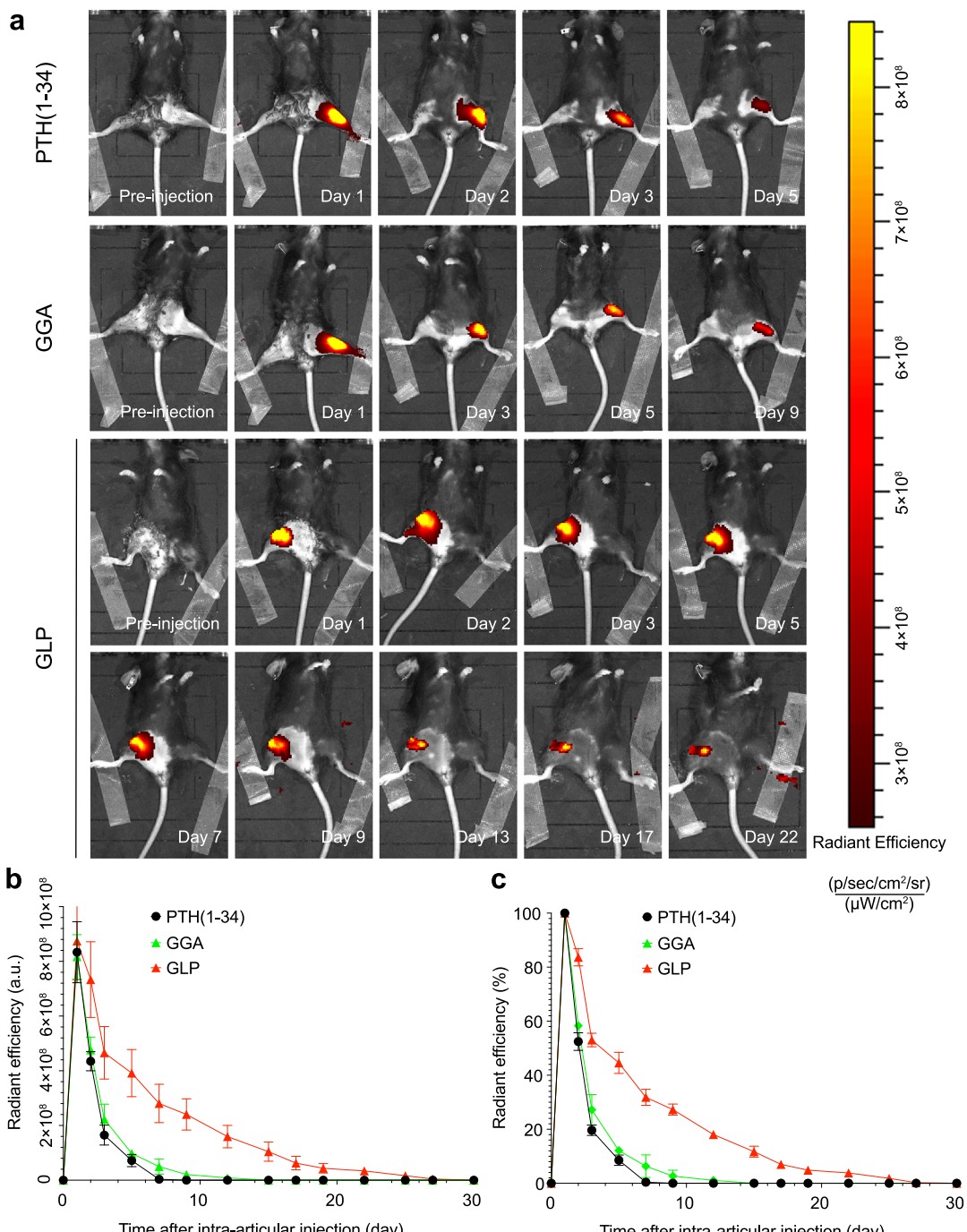

**Fig. 4 | The in vivo retention of hydrogel in arthrography. a** IVIS images for the mice after intra-articular injection with GLP hydrogel versus PTH (1–34) water solution with progress of time. **b, c** The radiant efficiency and radiant remained ratio of the samples after intra-articular injection calculated from IVIS imaging system. Data are presented as means ± SD of at least three replicate experiments. GGA gallic acid-grafted gelatin, GLP GGA@Lipo@PTH (1–34), A.u arbitrary units.

for all the mice with different treatments, indicating that intra-articular injection with hydrogels would not affect the knee motion of mice.

## In vivo retention of hydrogel in the joint cavity

The PTH (1–34), GGA, and GLP hydrogels were labeled with dye IR780 to evaluate the in vivo retention in the joint cavity with the IVIS Spectrum system. The IVIS results are shown in Fig. 4. From the IVIS images (Fig. 4a), after intra-articular injection, the radiant signals of the GLP hydrogel group gradually decreased, and the decrease rate of the radiant signals for such a group was slower than that of the GGA and PTH (1–34) control groups. The radiant signals for the GLP hydrogel group lasted for 22 days in the joint cavity, but only lasted 9 and 5 days in the GGA hydrogel and PTH (1–34) groups, respectively. The IVIS results strongly substantiated the radiant efficiency (a.u.) (Fig. 4b), and the residual radiant efficiency (%) could be detected and calculated (Fig. 4c). From Fig. 4b, the detected residual radiant efficiency for the GGA hydrogel and PTH (1–34) groups decreased faster than GLP hydrogel. After 5 and 9 days, only a few radiances were observed in the GGA hydrogel and PTH (1–34) groups, respectively, but the GLP hydrogel exhibited radiance for 22 days. The residual radiant efficiency (%) in Fig. 4c showed a similar tendency to that in in Fig. 4b.

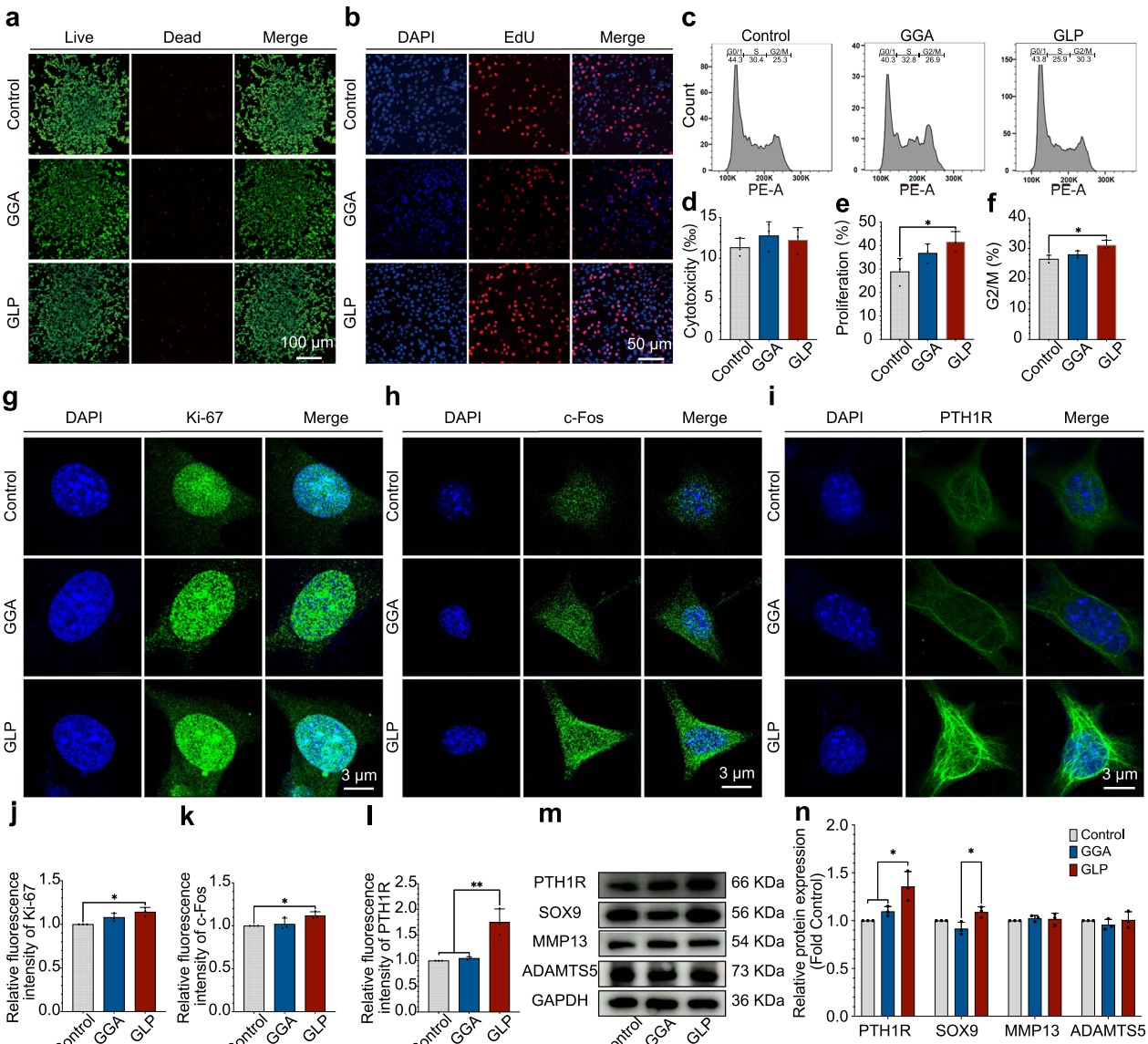

**Fig. 5 | The ATDC5 cells' behavior when cultured with GLP hydrogel. a, d** Live/dead stained fluorescent images and statistical analysis for cells' cytotoxicity, green stand for the live cells and red is the dead cells. **b, e** EdU/DAPI stained fluorescent images to evaluate the cell proliferation (red represents the newly proliferated cells (EdU staining), blue represents all cells (DAPI staining). **c, f** Flow cytometry (FCM) results and G2/M rate calculated from FCM results. **g–l** Immunofluorescent images for Ki-67, c-Fos, and PTH1R expression for the ATDC5 cells cultured with hydrogels and the statistical stained positively area for the expression of such genes. **m, n** Western Blot results for the expression of proliferation proteins (PTH1R, SOX9, MMP13, ADAMTS5) of ATDC5 cells when cultured with hydrogels and the corresponding statistical analysis results. Data are presented as means ± SD of at least three replicate experiments. Unpaired two-tailed Student's *t* tests was used to calculate significant difference, *$p < 0.05$, **$p < 0.01$. GGA gallic acid-grafted gelatin, GLP GGA@Lipo@PTH (1–34).

## In vitro ATDC5 cells behavior with hydrogels

Live/dead staining was utilized to evaluate the cytotoxicity of the hydrogels, and the results are presented in Fig. 5. As shown in Fig. 5a, a relatively small number of dead cells (red dots) were observed in the GLP hydrogel and control groups, and the statistical results in Fig. 5d confirmed there were no significant differences between the GLP hydrogel and control groups in terms of cytotoxicity. EdU staining was applied to evaluate the cell proliferation, and the results are shown in Fig. 5b, e. In the fluorescent images (Fig. 5b), the red dots (EdU-stained) are the newly proliferated cells, and the blue dots are the total cells. As shown, the newly proliferated cells for EdU-stained cells in GLP hydrogel were higher than those of the control groups, and the statistical analysis results in Fig. 5e confirmed such a tendency. The flow cytometry (FCM) results in Fig. 5c, f showed that the G2/M of GLP hydrogel was 30.3%, which was higher than the 25.3% in

the control group or the 26.9% in the GGA hydrogel group. The immunofluorescence staining results for the monoclonal antibodies Ki 67 (Ki-67), FBJ osteosarcoma oncogene (c-Fos), and parathyroid hormone 1 receptor (PTH1R) were presented in Fig. 5g–i. The Ki-67 and c-Fos proteins were expressed in the nucleus of ATDC5 cells, which are the cell proliferation and anti-apoptosis genes; PTH1R was expressed on the surface of ATDC5 cells, which is the receptor for PTH (1–34). From the images in Fig. 5g, h, the ATDC5 cells that were cultured with GLP hydrogel presented a brighter positive staining color for Ki-67, c-Fos, and PTH1R expression compared with the control groups, and the corresponding statistical analysis results in Fig. 5j, l confirmed the immunofluorescent observations.

The Western Blot results in Fig. 5m show that the blots for the PTH1R and SOX9 protein expression for the GLP hydrogel cultured ATDC5 cells were higher than those of the control groups, but the

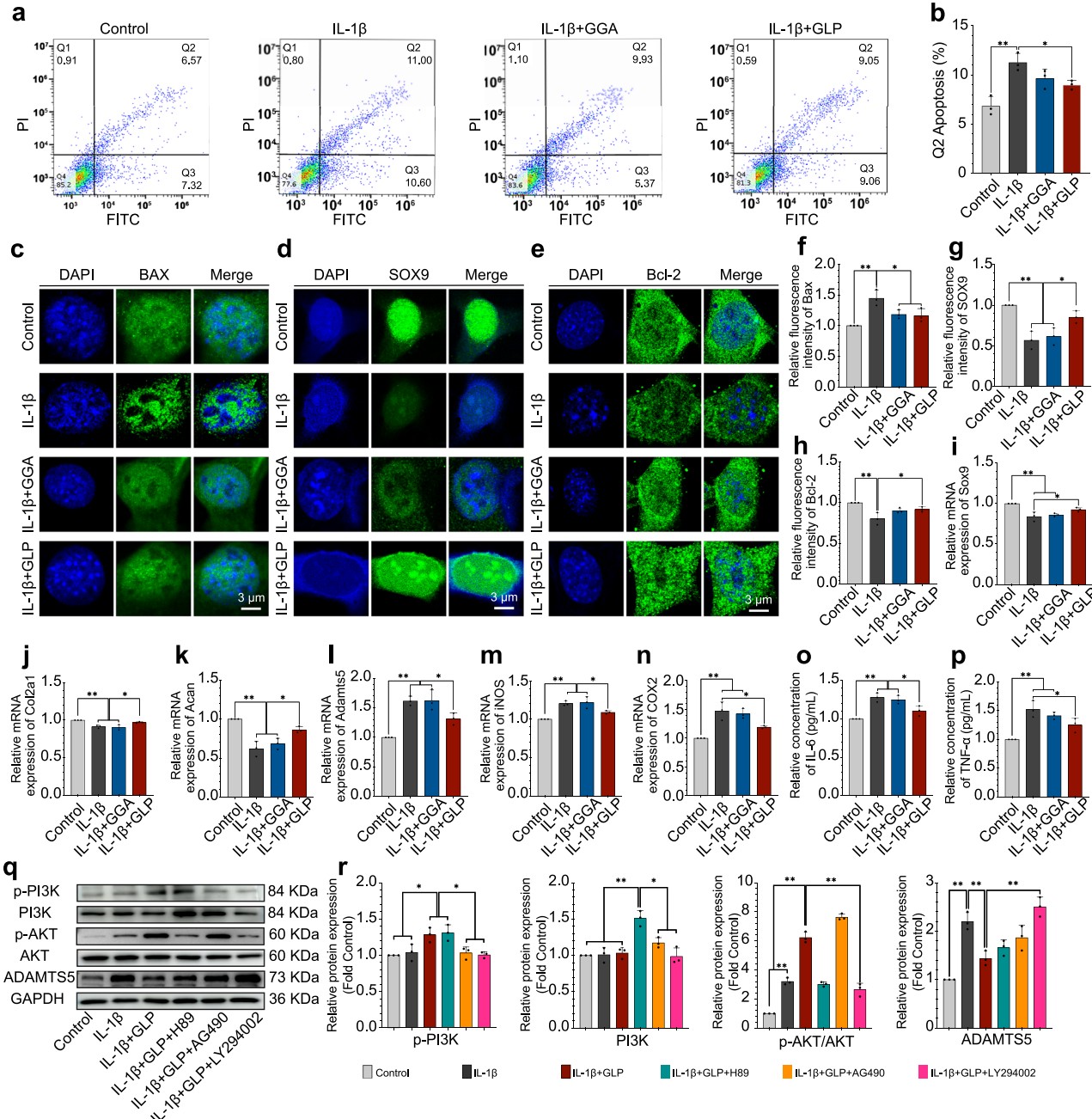

**Fig. 6 | IL-1β-induced ATDC5 cells' behavior cultured with GLP hydrogel compared with GGA hydrogel. a, b** The FCM results and statistical analysis results for the percentage of apoptosis. **c–h** Immunofluorescent images for SOX9, Bcl-2, and BAX protein expression as well as the statistical stained positively area from the images. Sox9 is the characteristic gene to indicate the ECM expression, Bcl-2 and Bax genes are the anti-apoptosis and the apoptosis-promoting genes of the cells. **i–n** The RT-qPCR results of the mRNA expression levels for Sox9, Col2a1, Acan, Adamts5, iNOS, and COX2. **o, p** The ELISA results for inflammatory mediators, IL-6 and TNF-α. **q, r** The Western Blot results for the expression of proteins (p-PI3K, PI3K, p-AKT/AKT, ADAMTS5) of IL-1β-induced ATDC5 cells cultured with hydrogels and the corresponding statistical analysis results. Data are presented as means ± SD of at least three replicate experiments. Unpaired two-tailed Student's *t* tests was used to calculate significant difference, \**p* < 0.05, \*\**p* < 0.01. GGA gallic acid-grafted gelatin, GLP GGA@Lipo@PTH (1–34).

blots for the matrix metallopeptidase 13 (MMP13) and ADAMTS5 proteins showed no significant differences between the different samples. The statistical results in Fig. 5n confirmed the observational results of the protein blots. The indicated GLP hydrogel could promote chondrocyte proliferation via upregulating the expression of the Ki-67 and c-Fos.

### In vitro IL-1β-induced ATDC5 cells behavior with hydrogels
IL-1β is a commonly used component in studies for stimulating the catabolic and inflammatory gene expression of chondrocytes, and

IL-1β-induced ATDC5 cells can be measured in isolated cells as a cell model to mimic the destructive environment of progressive OA[30]. Figure 6 showed the behavior of IL-1β-induced ATDC5 cells when cultured with GLP hydrogel versus GGA hydrogel. The FCM results in Fig. 6a show the percentage of Q2 (indicating the apoptosis) in ATDC5 cells. After being induced with IL-1β, the apoptosis rate was increased to 11.00%, but the parameter decreased to 9.93% after treatment with the GGA hydrogel and even decreased to 9.05% after culture with the GLP hydrogel. The apoptosis rate for the GLP group was close to the control group (6.57%) but the difference was not statistically

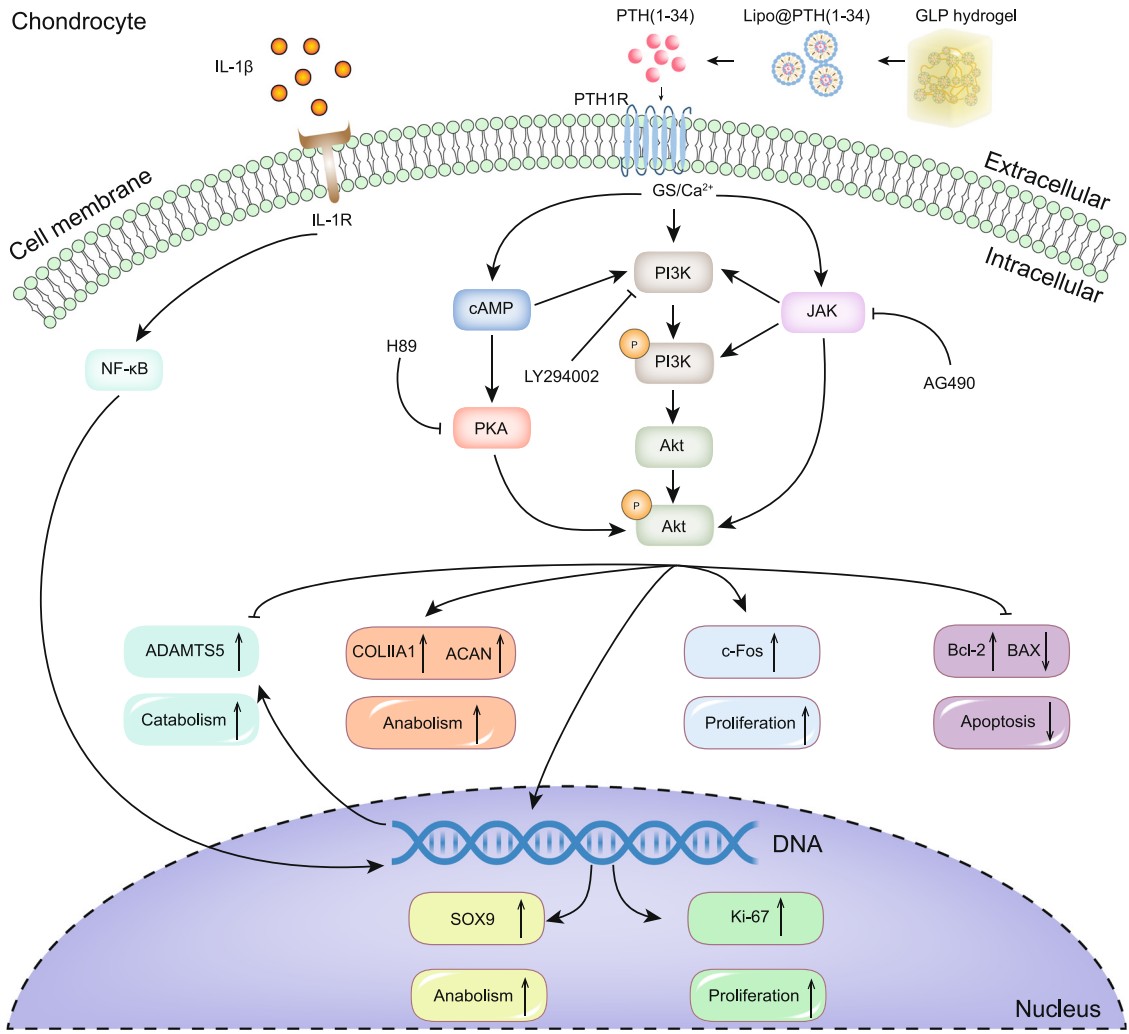

**Fig. 7 | The potential mechanism of the GLP hydrogel to ATDC5 cells.** The GLP hydrogel would promote ATDC5 cells proliferation by up-regulating the expression of cell proliferation and anti-apoptosis genes (Ki-67, c-Fos), and this hydrogel would protect IL-1β-induced ATDC5 cells from further progression by up-regulating the expression of key anabolic genes (Sox9, Bcl-2, Col2a1, Acan) and down-regulating the expression of key catabolic genes (Bax, Adamts5), which potentially suggested regulating the PI3K/AKT signaling pathway. GLP GGA@Lipo@PTH (1–34).

significant. The statistical analysis results in Fig. 6b showed the same tendency as the FCM results.

Figure 6c–e showed the immunofluorescence staining results for the BCL2-associated X protein (BAX), SOX9, and Bcl-2 protein expression. The Sox9 gene is a characteristic gene for indicating the ECM expression of cells and the expression of SOX9 indicates the secretion of ECM; The Bcl-2 gene is the anti-apoptosis gene of a cell; while the Bax gene is an indicator gene for the promotion of cell apoptosis. As shown in the immunofluorescence images, GLP hydrogel promoted a higher degree of SOX9 and Bcl-2 protein expression when compared with IL-1β-induced cells, but presented a lower level of BAX protein expression compared with the IL-1β-induced cells. Notably, the GGA hydrogel exhibited comparable expression in such genes versus the control group. The statistical results in Fig. 6f–h confirmed the immunofluorescent observations.

The quantitative reverse transcription polymerase chain reaction (RT-qPCR) results in Fig. 6j–n displayed that the GLP hydrogel exhibited higher levels of Sox9, Col2a1, and Acan gene expression when compared with the GGA hydrogel and IL-1β-treated control groups, and such upregulated genes were the anabolic markers. The expression levels of catabolic genes, Adamts5 and iNOS in the GLP hydrogel treated group were lower than those of the GGA hydrogel treated groups and the IL-1β-induced control groups.

The enzyme-like immunosorbent assay (ELISA) results for the expression of inflammatory mediators, interleukin 6 (IL-6), and tumor necrosis factor (TNF-α), in Fig. 6o, p, showed that the IL-6 and TNF-α expression levels for the GLP hydrogel treated group were lower than those of the GGA hydrogel treated group and the IL-1β treated control group. Such results indicated the potential anti-inflammatory effect of the GLP hydrogel.

The Western Blot results in Fig. 6q, r showed the expression levels of relevant key proteins in the PI3K/AKT signaling pathway when the IL-1β-induced ATDC5 cells were co-cultured with the GLP hydrogel with or without different inhibitors. The potential mechanism is presented in Fig. 7. The expression levels of key proteins, p-PI3K, PI3K, p-AKT, and AKT in the PI3K/AKT signaling pathway were investigated. As shown in Fig. 6q, the GLP treated group showed a higher p-PI3K protein expression as compared with the IL-1β-induced and control groups, and with the addition of a JAK2 inhibitor (AG490) or a PI3K inhibitor (LY294002), the p-PI3K protein expression was decreased, but the parameter did not change with the presence of a PKA inhibitor (H89). Such findings verify that the GLP hydrogel might activate the p-PI3K protein expression (as shown in Fig. 7). From the PI3K blot, the GLP hydrogel treated group showed no difference in PI3K protein expression when compared with the IL-1β induced and control groups, but with the addition of H89 or AG490, the PI3K protein expression

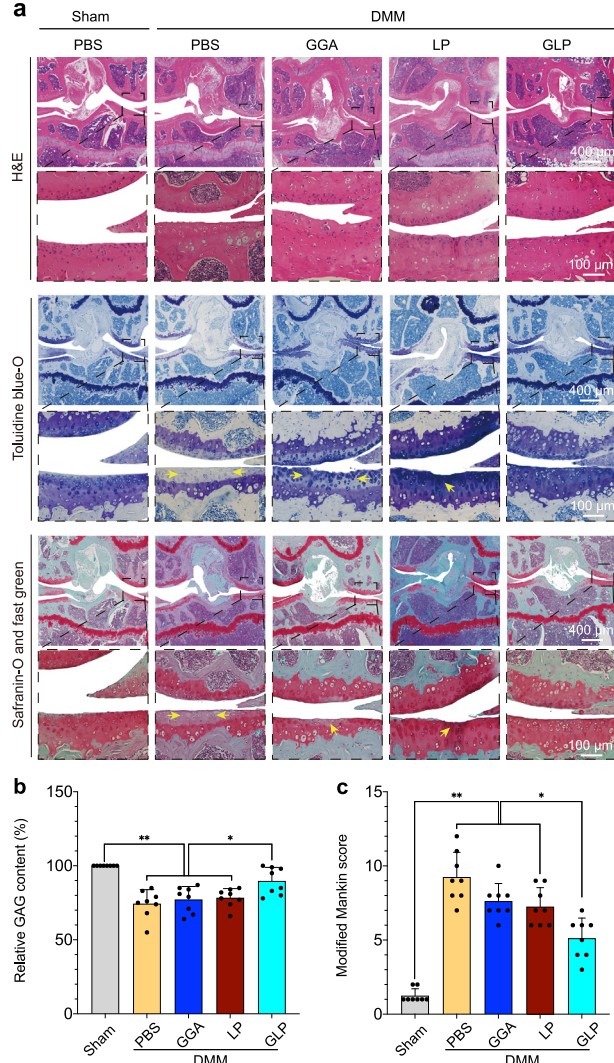

**Fig. 8 | Section staining for the arthrography area after intra-articular injection with the GLP hydrogel. a** H&E, Toluidine blue-O, and Safranin-O and fast green staining. The yellow arrows in these stained images represent the degenerated areas in the articular cartilage. **b** Statistical analysis for the relative GAG content calculated from Toluidine blue-O and Safranin-O and fast green stains. **c** Modified Mankin scores calculated from the stains of the described sections according to the Modified Mankin scores in Supplementary Table S2. Data are presented as means ± SD ($n = 8$). Unpaired two-tailed Student's $t$ tests was used to calculate significant difference, *$p < 0.05$, **$p < 0.01$. DMM destabilization of the medial meniscus, PBS phosphate-buffered saline, GGA gallic acid-grafted gelatin, LP Lipo@PTH(1–34), GLP GGA@Lipo@PTH (1–34).

protein expression was further increased again when compared with the group treated solely with GLP hydrogel. Such findings indicated that the GLP hydrogel inhibited the degradation of the ECM. The statistical analyses for such protein expressions in Fig. 6r confirmed the blots observation. The results also verify that the GLP hydrogel is a significant factor in protecting IL-1β-induced-ATDC5 cells from further OA progression, potentially through the PI3K/AKT signaling pathway.

In addition, a DCFH-DA assay was performed to detect the ROS level, and the positively stained fluorescent images are displayed in Supplementary Fig. 8. A higher number of positively stained cells indicated a higher ROS level. As shown in Supplementary Fig. 8a, the IL-1β-induced sample showed a higher number of positively stained cells compared with the control group, but after being cultured with GGA and GLP hydrogels, the positively stained cell number decreased, especially for the GLP hydrogel treated group. The quantitative statistical analysis results in Supplementary Fig. 8b confirmed the immunofluorescent observation results.

### In vivo biosafety of GLP hydrogel after intra-articular injection in DMM mice
After GLP hydrogel was intra-articularly injected in DMM mice for 10 weeks, the key organs, including the heart, liver, spleen, lung, and kidney, were obtained for hematoxylin-eosin (H&E) staining, and the results are shown in Supplementary Fig. 9. From the H&E stained images, the histomorphology of the various organs notably demonstrated no significant difference when compared. Such a characteristic nature indicated the non-toxic behavior of intra-articular injection of the GLP hydrogel into the main organs of the tested mice.

### In vivo OA treatment effect of GLP hydrogel
**Micro-CT evaluation of the tibia plateau.** The macrographic images and the Micro-CT data are presented in Supplementary Fig. 10. According to Supplementary Fig. 10a, the GLP hydrogel treated surface of the tibia plateaus exhibited a lesser wear or tear trace compared with the other groups (the black arrow represents the wear and tear areas), while the three-dimension images of Micro-CT in Supplementary Fig. 10b also exhibited the same tendency. The region of interest (ROI) area for Micro-CT analysis is shown in Supplementary Fig. 10c, d. The bone parameters were provided in Supplementary Fig. 10e–g, and the results showed that there were no significant differences among the samples in all the bone parameters.

**Histological evaluation.** H&E staining, toluidine blue-O staining, and safranin-O fast green staining assessments were performed to investigate the effect of the GLP hydrogel on the cartilage, since the cartilage is sensitive to the progression of OA. The results are shown in Fig. 8. The thickness of the cartilage and smoothness of the surface (presented in the magnified images) indicate the degree of degradation the cartilage; a thicker and smoother cartilage indicat lesser degradation of the cartilage in OA. At the same time, the uniformity of the stained cartilage tissue is also one of the essential indicators for evaluating the degradation degree of cartilage in the OA model. More uniformly stained cartilage indicated lesser degradation of the cartilage. In the results, the sham group included normal mice without the DMM surgery, which served as a positive control, and DMM mice that were injected with PBS served as a negative control. From the histological staining results in Fig. 8a, for H&E staining, the DMM group with PBS presented non-uniform red-stained tissue (yellow arrows), and the red-stained tissues were more homogeneous with injection of GGA hydrogel, Lipo@PTH (1–34), and GLP hydrogel. Among the DMM groups, the GLP hydrogel treated group presented the most homogeneous, red-stained tissue, indicating the least degradation of cartilage. The homogeneity of the stained tissue for the GLP hydrogel was even comparable to the sham group. The toluidine blue-O and safranin-O fast green stainings also exhibited the same tendency as the

was increased. At the same time, with the presence of the LY294002, the PI3K protein expression decreased, which indicated that the GLP hydrogel would also activate the P13K protein expression. From the p-AKT blot, the GLP hydrogel treated group presented higher levels of p-AKT protein expression compared with the IL-1β treated group and control group. However, with the addition of the LY294002, the p-AKT protein expression decreased, but there was no difference versus the sole GLP group with the addition of the AG490. Such findings indicated that the GLP hydrogel would activate p-AKT protein expression. The expression level of the catabolic marker protein ADAMTS5 was also investigated. The catabolic marker protein ADAMTS5 blot in Fig. 6q showed that the ADAMTS5 protein expression IL-1β-induced group was higher than that of the control group, but the ADAMTS5 protein expression was decreased after being treatment with the GLP hydrogel. Further, with the addition of the LY294002, the ADAMTS5

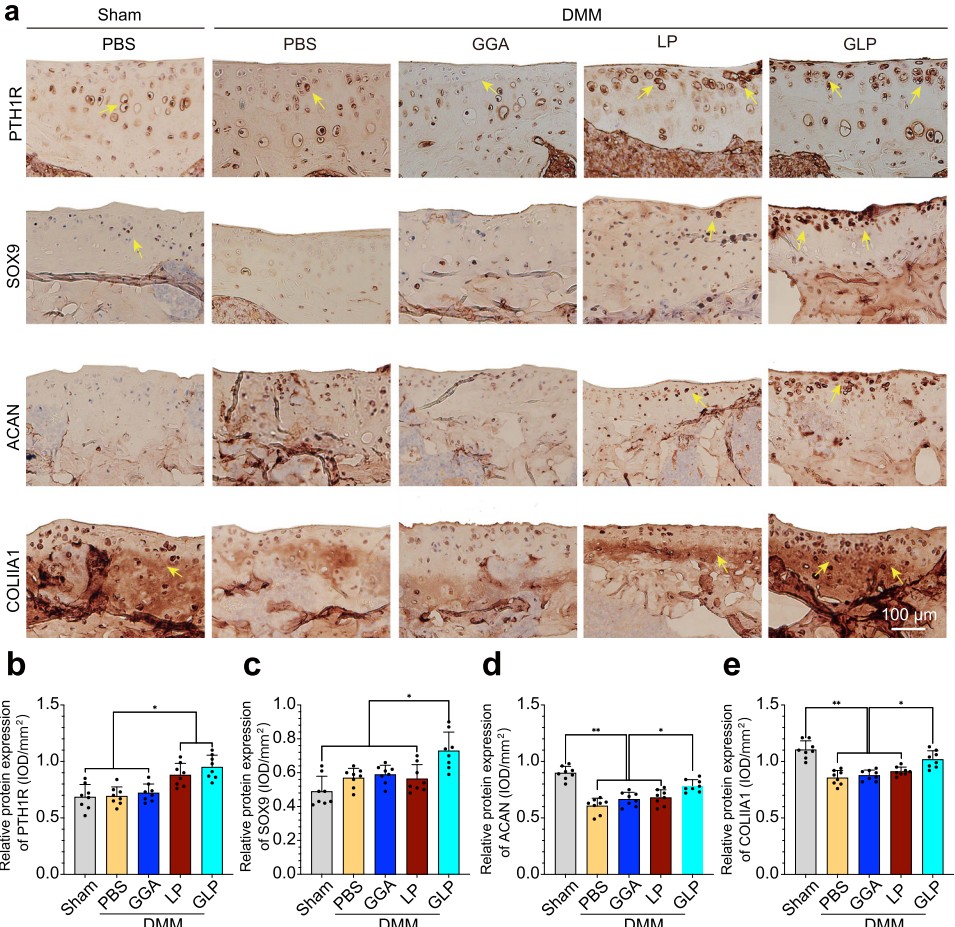

**Fig. 9 | Representative immunohistochemical staining of the cartilage-related markers in the cartilage after intra-articular injection with GLP hydrogel.**
**a** Immunohistochemical stained images for PTH1R, SOX9, ACAN, and COLIIA1 proteins. The yellow arrows stand for the immunohistochemical positively stained areas in the articular cartilage. **b**–**e** The statistical analysis results for the expression of these proteins from immunohistochemical stained images. Data are presented as means ± SD ($n = 8$). Unpaired two-tailed Student's $t$ tests was used to calculate significant difference, *$p < 0.05$, **$p < 0.01$. DMM destabilization of the medial meniscus, PBS phosphate-buffered saline, GGA gallic acid-grafted gelatin, LP Lipo@PTH(1–34), GLP GGA@Lipo@PTH (1–34).

H&E staining results in terms of the homogeneity of stained tissue. Aside from the homogeneity, the toluidine blue-O and safranin-O fast green stainings can reflect the thickness of stained cartilage. The blue-stained area in toluidine blue-O staining and the bright red-stained area in safranin-O fast green staining represent the stained cartilage, which reflected the thickness of the cartilage. Such results also display the same tendency as H&E staining. Compared with the non-uniform stained areas of PBS, GGA hydrogel, and Lipo@PTH (1–34) treated groups, the GLP hydrogel treated group presented a more homogenous positively stained cartilage layer. More uniform staining in the cartilage layer indicates a lesser degree of degradation of the cartilage. The Masson and Alcian blue staining results in Supplementary Fig. 11 showed the same tendency. GAG is the main component of ECM in the cartilage. According to the present staining results, the assumed content of the GAG versus the sham group was calculated and the results are presented in Fig. 8b. The results confirmed the aforementioned knee staining results. In DMM mice, the GLP hydrogel treated group presented the largest GAG-assumed content. Moreover, the severity of the joints was evaluated according to the staining results with the principal Modified Mankin score[31] shown in Supplementary Table 1, and the results are presented in Fig. 8c. In DMM mice, the GLP hydrogel treated group presented the lowest score.

**Immunohistochemical staining.** Immunohistochemical staining for the expression of PTH1R, SOX9, ACAN, and COLIIA1 proteins was investigated in the cartilage area after 10 weeks of intra-articular injection, and the results are shown in Fig. 9. From the staining results (Fig. 9a), in DMM mice, the GLP treated group and the Lipo@PTH (1–34) treated hydrogel group exhibited a dark-brown stained color versus a light-brown stained color for the PBS and GGA treated groups in the PTH1R immunochemical staining images (yellow arrows). However, in all other immunochemical staining results, the GLP hydrogel treated group exhibited a dark-brown stained color versus a light-brown stained color for the other samples. The statistical results shown in Fig. 9b–e confirmed the results of the immunochemical staining images. The tissue staining results demonstrated that the GLP hydrogel could alleviate the OA progression by GAG secretion as well as protect against the degradation of cartilage in OA situations. Additionally, several anabolic and catabolic biomarkers, including MMP13, ADAMTS5, Bcl-2, and BAX proteins were also investigated and the results are presented in Supplementary Fig. 12, which could reversely reflect the ROS-scavenging property of the present hydrogel system as well as the ability to alleviate the progress of OA. From the immunohistochemical staining images, lesser stained brown dots in the cartilage layer (positively stained area) were presented in the expression of MMP13 and BAX proteins for the GGA and GLP hydrogel treated groups when compared with the PBS-treated DMM group. Further, among hydrogels treated groups, the GLP hydrogel-treated groups presented the least stained brown area. The statistical analysis results in Supplementary Fig. 12b–e confirmed the immunohistochemical staining observations.

## Discussion

OA is a seriously degenerative disease, especially in the elderly, that causes considerable pain to affect patients. OA will cause the degradation of chondrocytes and ultimately lead to surface fibrillation of the cartilage layer or even to cartilage defects[32]. As aforementioned, systematic or topical use of analgesics or anti-inflammatory drugs is used in clinical treatment. Topical intra-articular injection of drugs is an effective method, but suffers from rapid loss and frequent injections.

As such, injectable biomaterials with drug loading and controlled release properties, especially natural polysaccharides-based injectable hydrogels, have attracted significant attention for OA treatment[33]. In the present study, a liposome loaded PTH (1–34)-incorporated gallic acid-grafted gelatin injectable hydrogel was synthesized by means of a mild and nontoxic enzymatic method. Such a method is beneficial for protecting the bioactivity of biomolecules, such as certain vulnerable proteins or drugs. PTH (1–34) is water-soluble and its aqueous solution is easily diffused in the joint cavity. Therefore, in the present hydrogel system, water-soluble PTH (1–34) was first encapsulated with hydrophobic bilayer liposomes to control the release, and then the liposomes loaded with PTH (1–34) were further incorporated into the GLP hydrogel, which guarantees the controlled release property of PTH (1–34). In the hydrogel formation, the TG enzyme, an essential enzyme in the human body, was utilized as a type of biocatalyst to promote the synthesis of covalent isopeptide bonds between peptide-bound glutamine and lysine residues in GGA[34], resulting in the formation of the GGA hydrogel. Such enzymes had high activity at a certain temperature in the physiological environment (37 °C), and the addition amount of the enzyme and the temperature would regulate the GGA gel formation time. In a certain TG amount (5 wt% in the present case), the hydrogel would be formed in ca. 5 min at 37 °C. Once the hydrogel was formed, the hydrogel state would be stable at a different temperature, which is different from gelatin sol-gel transformation at a different temperature. When compared with recent reported injectable hydrogel systems[35,36] for OA treatment, the fabrication process of the present GLP hydrogel was mild, nontoxic, and there was no need for complicated polymerization and/or other harsh conditions. Thus, the present hydrogel system is desirable for clinical use and industry. Notably, the loading of PTH (1–34) into liposomes and the amount of Lipo@PTH (1–34) nanoparticles into the hydrogel system were optimized based on the loading efficacy of PTH (1–34) in liposomes, the in vivo retention of hydrogel in the joint cavity, and the effective concentration of PTH (1–34) for OA treatment.

In addition to being a mild, simple, and nontoxic synthesis method, the present GLP hydrogel possesses other attractive properties for OA treatment, including swelling restriction in DI water and PBS solution, gradual degradation behavior, and no non-biodegradable components. The swelling property is essential for hydrogels in tissue engineering, and most hydrogels, especially those with a large amount of amino groups (such as polyacrylamide), will swell in an aqueous solution, resulting in hydrogel expansion. The expanded hydrogels would weaken the mechanical properties and oppress the surrounding tissue at the implanted site, thereby impeding tissue regeneration. Therefore, a swelling restriction property for hydrogels is also attractive, especially for the joint cavity[37]. The GLP hydrogel is restricted to swelling in aqueous solutions, which could be attributed to the main components in gelatin being proteins, which usually have the same or similar amount of hydrophilic amino groups and hydrophobic carboxyl groups. In addition, when immersed in an aqueous solution, the formed protonated $NH_3^+$ would combine with the carboxyl group in gelatin, weakening the interaction between water and amino groups and thus suppressing the penetration of water into the hydrogel network[38]. In the present GLP hydrogel system, there are no non-degradable components, which endows the hydrogel with a full-biodegradable property. The gradual degradation of the GLP hydrogel in PBS for 28 days (Supplementary Fig. 4) could be attributed to the hydrolysis of proteins in gelatin and TG-induced combination in the GLP hydrogel. Notably, in the present hydrogel system, only a small amount of liposomes, especially incorporated in the outer layer, would be directly released from hydrogel. However, most of the liposomes in the inner hydrogel would not release from the hydrogel, since such inference was indirectly confirmed by the gradual release of PTH (1–34) from the GLP hydrogel (in vivo retention result in Fig. 4). In the present study, in vivo retention by intra-articular injection of GLP hydrogel was performed to investigate the PTH (1–34) release property, which is more suitable for investigating the release property of PTH (1–34) in the GLP hydrogel compared with an in vitro immersion study. In the pilot study, in vitro, the PTH (1–34) release profile of GLP hydrogel was investigated by immersing the GLP hydrogel in PBS solution at 37 °C for 28 days, but the PTH (1–34) in the GLP hydrogel immersed PBS solution was not detected using the HPLC-MS method. This might be attributed to the degradation products from GLP hydrogel potentially affecting the HPLC-MS detection. In this hydrogel system, the release of PTH (1–34) from the hydrogel was mainly through diffusion due to the concentration difference, and therefore the PTH (1–34) was released fast in the initial stages (in the first 3–5 days) before decreasing. In the process of PTH (1–34) release, the PTH (1–34) that diffused out from the hydrophobic liposome bilayer would be regarded as a release rate determining step, and the PTH (1–34) would release from the hydrogel easily through solution exchange after diffusing from the liposomes. Note that the IVIS method was used to evaluate the release of PTH (1–34) in this hydrogel system, which would be an alternative method to evaluate the release profile of PTH (1–34). As expected, different amounts of Lipo@PTH (1–34) incorporated into hydrogels presented different PTH (1–34) release concentrations and durations. In the pilot study, the ATDC5 cells cytotoxicity study with the different amounts of Lipo@PTH (1–34) incorporated hydrogels was performed, and the 1 wt% Lipo@PTH (1–34) incorporated hydrogel showed the cytotoxicity to the cells. Therefore, 0.5 wt% Lipo@PTH (1–34) incorporated hydrogel was used for the further study.

In the present GLP hydrogel system, the controlled release of PTH (1–34) guaranteed the bio-functionalities; the in vitro and in vivo studies confirmed such bioactivity. The GLP hydrogel showed no toxicity to the ATDC5 cells and promoted the proliferation of ATDC5 cells via upregulating SOX9 and PTH1R while down-regulating MMP13 and ADAMTS5 at the protein level. The GLP hydrogel could downregulate the catabolic genes, activate the anabolic genes, and protect the IL-1β-induced ATDC5 cells from further OA progression potentially through PI3K/AKT signaling pathway (as shown in Fig. 7). Such findings are notable for OA treatment in clinical practice. As is well known, the chondrocytes in OA patients are in an abnormal state, and the situation will worsen over time until the cartilage becomes defective. However, the GLP hydrogel would protect the normal function of chondrocytes even if such cells were in an inflammation state, while also promote the proliferation of the chondrocytes. Such findings differ from those reported in previous studies[39,40], in which more attention was paid to the promotion of chondrogenic differentiation in the stem cells. Such process is difficult since in most cases the chondrogenic induced stem cells ultimately promote the formation of fibrous cartilage (not the hyaline cartilage in normal arthrosis). Further, the in vivo intra-articular injection results confirmed that the GLP hydrogel not only protected the cartilage from degradation in DMM mice, but also promoted the synthesis of ECM.

In the present study, an indirect cell culture method was used to evaluate the cell behavior with hydrogels, which differs from the commonly used extraction method. The hydrogel samples were placed in a transwell and the cells were seeded on the bottom of the plate. The bioactive components released from the hydrogel would interact with the cells in real-time, but in the commonly used extraction method, the extraction involves the cumulative release of

bioactive components of samples after several days of immersion. As such, the belief of the present authors is that the cell culture method in the present study is more suitable for stimulating the actual situation of injectable hydrogels in the joint cavity. In the OA treatment, the injectable hydrogels in the joint cavity have the functions of drug loading and controlled released, but not the functions of scaffolding or filling the role for implanted materials which need to support the tissue ingrowth, such as hydrogel implantation in the bone defect area. Notably, the volume size of the hydrogel placed in the transwell and the medium volume in each well were optimized in the pilot study. Besides, the reasons for the in vivo intra-articular injection of hydrogel at 1 week after modeling were based on the following considereations. First, the surgical incision healing for DMM would be stable after 1 week. Second, it will take a quite long time to have significant OA symptoms after DDM surgery, which will bring more painfulness to the animals and increase the costs. Last but not the least, as OA is an irreversible disease, it may be hard to obtain a remarkable effect to retard the OA progression when it has significant OA symptons, since the primary design of this hydrogel was to alleviate OA progress and protect chondrocytes degradation.

Few limitations need to be addressed in the present report for further research progress. First, although there are no non-degradable components in the present hydrogel system, no fully in vivo biodegradable data was obtained. In situ in vivo retention of hydrogel data confirmed the gradual quenching of fluorescence in the hydrogel, but not hydrogel degradation. Second, the ATDC5 chondrogenic cell line was used in the present study as they are more stable than the human chondrocytes, especially for signaling pathway research. Human chondrocytes will be considered for future research, so as to be more clinically relevant. Third, in the present study, a strategy for loading and controlled release of water-soluble bioactive components in OA treatment was established, but the optimized intra-articular injection frequency is still every three weeks (although the frequency is lower than many reported studies), and therefore, the loading and controlled release efficiency will be a significant research direction in the future. Conclusively, the present study provides significant results on the protection of chondrocytes from OA and alleviation of OA progress; however, total reversal of OA is difficult to achieve.

In conclusion, PTH (1–34) was successfully encapsulated into liposomes, and a liposome loading PTH (1–34) incorporated gallic acid-grafted gelatin injectable hydrogel was effectively fabricated by means of a mild and nontoxic enzymatic-linking method. The GLP hydrogel was restricted to swell in PBS and DI water, and degraded gradually in PBS solution. The lyophilized GLP hydrogel presented a porous structure. The results of in situ colloidal formation of the hydrogels confirmed that the GLP hydrogel would be formed via intra-articular injection in the mice joint cavity, and the gait analysis showd that the knee motion would not be affected after intra-articular injection with hydrogels. The in vivo in situ retention results confirmed that the PTH (1–34) was released in a sustained manner from the GLP hydrogel, and the release time would last for over 20 days, but the PTH (1–34) in the aqueous solution in such setting would only last for 5 days. The GLP hydrogel would promote the proliferation of ATDC5 cells by up-regulating the expression of key proteins, Ki-67, c-Fos, and PTH1R. Also, the GLP hydrogel would protect the IL-1β-induced ATDC5 cells from further progression by up-regulating the expression of key anabolic proteins, SOX9, Bcl-2, COLIIA1, ACAN, and simultaneously down-regulating the expression of key catabolic or inflammatory proteins, BAX, ADAMTS5, iNOS, COX2, IL-6, and TNF-α, which potentially suggested regulating the PI3K/AKT signaling pathway. The results of in vivo intra-articular injection in DMM mice confirmed that the GLP hydrogel would protect the cartilage layer from degradation, alleviate OA progression, and promote the synthesis of GAG. Finally, the GLP hydrogel would not cause systemic toxicity. Overall, the GLP injectable hydrogel shows great potential in OA treatment.

## Methods

### Preparation and characterization of Lipo@PTH (1–34)
Lipo@PTH (1–34) was prepared according to previous studies[41]. Briefly, 20 mg cholesterol (Cat#C8667, Merck, Germany) and 80 mg lecithin (Cat#R-D0010, RuixiBio, China) were completely dissolved in 30 mL of chloroform in a round-bottom flask, then the chloroform was removed by means of evaporation at 35 °C for 1 h. Subsequently, 3 mL of DI water dissolved with different amounts, including 1.0, 1.5, or 2.0 mg PTH (1–34) (Cat# HY-P0059, MCE, US) was added to the flask, and then the flask was subjected to sonication at 37 °C for 1 h, forming micron-sized multi-lamellar liposomes. The different input amounts of PTH (1–34) were used to optimize the loading rate of PTH (1–34) in liposomes. To further fabricate smaller sized liposomes, the solution in the flask was treated with probe-sonication (JY92-IIN, Scientz, China) for 5 min (60 pulses/min, 130 W). Then, the solution in the flask was dialyzed in DI water for 24 h to remove the unencapsulated PTH (1–34). The Lipo@PTH (1–34) nanoparticles were collected and stored at 4 °C for further use. The liposomes without PTH (1–34) served as a control. The loading content of PTH (1–34) in Lipo@PTH (1–34) was determined by HPLC-MS (Thermo Exploris 480, US). For the HPLC-MS test, a concentration gradient curve for PTH (1–34) dissolved in DI water was first established; the PTH (1–34) concentration in the Lipo@PTH (1–34) dialyzed DI water was detected, and the PTH (1–34) loading efficiency in Lipo@PTH (1–34) was calculated according to the following equation: $(M_t - M_n)/M_t \times 100\%$, where $M_t$ is the total amount of PTH (1–34) added to prepare Lipo@PTH (1–34) nanoparticles, and $M_n$ is related to the non-loaded PTH (1–34) amount detected in dialyzed DI water. The liposomes without PTH (1–34) were fabricated by the same method, which served as the control.

The morphology of Lipo@PTH (1–34) nanoparticles was observed by means of a TEM (Hitachi, HT7800, Japan) and the samples were negatively stained with 2% phosphotungstic acid. The average size, polydispersity index, and Zeta potential of Lipo@PTH (1–34) were determined by means of Dynamic Light Scattering (Zeta-sizer Nano S90, Malvern Instruments, Worcestershire, UK).

### Preparation of GGA
GGA polymers were synthesized by following the protocol in a previous study[27]. Briefly, 5 g gelatin (Cat#V900863, Sigma, US) was dissolved in 150 mL DI water using a water bath at 40 °C. Simultaneously, 3.4 g gallic acid (GA, Cat#G823163, Macklin, China) (20 mmol) was dissolved in 125 mL DI water and N,N-dimethylformamide (DMF, Aladdin, China) mixed solution (the volume ratio for DI water and DMF was 3:2), and then 3.82 g N-(3-Dimethylaminopropyl)-N'-ethylcarbodiimide hydrochloride (EDC, Cat#N808856, Macklin, China) and 3.2 g N-Hydroxysuccinimide (NHS, Cat#H6231, Macklin, China) were added into the GA solution to activate the carboxylic groups of GA. After the GA mixture was stirred for 1 h at RT, the GA solution was added to the gelatin solution, and the mixture was stirred overnight at 40 °C to complete the reaction. Subsequently, the reacted solution was dialyzed in DI water for one week, and the DI water was changed twice per day. The dialysis solution was then filtered and freeze-dried to obtain the GGA polymer. NMR spectroscopy (Bruker Avance 300 MHz NMR spectrometer) and fourier transformed infrared (FTIR) spectra (Thermo Nicolet iS10) were utilized to analyze the chemical structures.

### Fabrication of the GLP hydrogel
The synthesized GGA polymers were dissolved at 10 wt% in DI water at 50 °C, and then the solution was cooled down to 37 °C. The 0.5 wt% of Lipo@PTH (1–34) nanoparticles was added to the GGA solution and after mixing well, 0.5 wt% of TG (Cat#S10156, Shanghai yuanye Bio-Technology, China) was dissolved in the mixed GGA solution, followed by incubation at 37 °C for ca. 5 min to form the hydrogel. The hydrogel was marked as GGA@Lipo@PTH (1–34) (GLP hydrogel). For the control, 0.5 wt% of TG was added into GGA solution (without any other

components) and incubated at 37 °C for ca. 5 min to form the hydrogel, which was marked as GGA hydrogel. The different concentrations of Lipo@PTH (1–34) incorporated into the hydrogel were utilized to assess the PTH (1–34) release curves. For this assessment, 2.0 mg PTH (1–34) and 0.23 mg Fluorescein 5-isothiocyanate (FITC) (Cat#AT101C, Multi Sciences, China) were dissolved in a 2 mL dimethyl sulfoxide (DMSO, Cat#D806645, Macklin, China) solution, and after reation at room temperature (RT) for 24 h, the FITC-labeled PTH (1–34) was synthesized. Then, the reacted solution was dialyzed (the molecular weight of the dialysis bag was 14 kDa, Sigma, US) in DI water for three days to remove the non-reacted FITC and DMSO solvent. During the dialysis process, the PTH (1–34) would not diffuse out of dialysis bag. Finally, a 3 mL FITC-labeled PTH (1–34) DI water solution was obtained. After that, the FITC-labeled PTH (1–34) was utilized for the synthesis of FITC-labeled Lipo@PTH (1–34) nanoparticles following the above-mentioned process. Then, different amounts of FITC-labeled Lipo@PTH (1–34) nanoparticles, including 0.125, 0.25, 0.5, and 1 wt%, were respectively added into a 10 wt% GGA solution; after including 5 wt% of TG 200 μL of each kind of FITC-labeled Lipo@PTH (1–34) mixed pre-hydrogel solutions was added into each well in a 96-well plate. After the formation of the hydrogel, 200 μL of PBS solution was added into each well and incubated at RT to assess the PTH (1–34) release profile. At the set time points, the PBS in each well was transferred to a new 96-well plate, followed by the IVIS detection of these immersed PBS solutions and hydrogels; after that, 200 μL of fresh PBS was added to each well until the next set time point. Then, the released PTH (1–34) in PBS solution was calculated from the standard curves of the concentration gradient of FITC-labeled PTH (1–34) in DI water according to the IVIS detection. Finally, the IVIS images for hydrogels and the accumulated PTH (1–34) release profile were obtained.

## Characterization

The molecular structure for the synthesized GGA polymers was detected by means of $^1$H NMR. A digital camera was used to image the hydrogels. The dynamic rheological tests of hydrogels were conducted with a Rheometric Scientific HAAKE (Malvern, UK), and the tests were performed at 37 °C. After 270 μL of the GLP mixture was mixed with TG solution, the mixed solution was immediately placed onto a quartz plate (20 mm in diameter and 1 mm in height) of the rheometer. Then, a time sweep was performed with a strain of 1%. For the frequency scanning, the formed disk hydrogel (20 mm in diameter and 1 mm in height) was placed on the plate and the frequency was scanned from 0.1 to 10 Hz at a strain of 1%.

The surface morphology for the lyophilized hydrogels was imaged by means of SEM (MERA3 TESCAN). The pre-hydrogel solution was stored in a 2.5 mL syringe, and after hydrogel formation, the hydrogel was cut into a disk sample (8 mm in diameter and 2 mm in height), followed by lyophilization. After gold sputtering, the surface morphology was observed under SEM.

The chemical composition of lyophilized hydrogels was detected by means of FTIR and XPS (Thermo Fisher K-Alpha, US) equipped with a monochromatic Al Kα (1486.6 eV) at 12 kV × 15 mV under the pressure of $2 \times 10^{-7}$ Pa.

## Swelling ratio and degradation behavior of hydrogels

The fabricated cylindrical hydrogels (approximately 8 mm in diameter and 10 mm in height) were prepared to investigate the swelling ratio. The macroscopic photoes of the as fabricated hydrogels were imaged, and after swollen in DI water or PBS solution (37 °C), the macroscopic photoes of hydrogels were taken again.

The fabricated hydrogels were cut into cylinders (8 mm in diameter and 10 mm in height) and then immersed in PBS solution in a rocking state for up to 28 days. The immersed PBS was changed every three days. Before the immersion, the hydrogels were lyophilized and weighted as $W_0$; at each set time point, the hydrogels were lyophilized

and weighted as $W_1$. The mass remaining ratio of hydrogels was calculated as $W_1/W_0 \times 100\%$.

## Mechanical property tests of hydrogels

The hydrogels were formed in a man-made mould or syringe to fabricate the rectangular or cylindrical samples for mechanical property tests. For the rotation test, the rectangular samples (100 mm × 10 mm × 1 mm) were rotated for 360° at RT for 30 s. The same size of rectangular samples was used for stretching tests, which were performed with a static and dynamic material testing machine (Moxin electronic tension machine MX-0580, Jiangsu, China) with a stretching speed of 20 mm/min. The cylindrical samples (10 mm in heigh and 8 mm in diameter) were used for compression testing using the same material testing machine with a compression speed of 5 mm/min.

## The in situ colloidal formation of hydrogel in vivo

All mice (female, 8-week-old) were purchased from Shanghai Model Organisms (Shanghai, China) and housed in Topbio-technology (Shenzhen, China) under standard laboratory conditions (12 h light/dark cycle, ambient temperature (25 °C), and humidity (50%)) with food and water freely. Six mice were randomly divided into two groups ($n = 3$). All the animal procedures were conducted following the instructions approved by the Institutional Animal Care and Use Committee (IACUC) of Peking University Shenzhen Hospital. All the animal study used this kind of mice and the animal treatment process was followed this IACUC protocol (No.2021-501). After anesthesia with isoflurane, the hairs around the knee were shaved. Then, the GLP pre-hydrogel solution and PBS (served as a control) mixed with Carbon Nanoparticles Suspension Injection (LUMMY, China) was respectively intra-articularly injected in mice. After 30 min, the mice were sacrificed and the anatomy of the joint cavity was identified to observe the colloidal formation of the GLP hydrogel.

## Gait analysis of mice after intra-articular injection with GLP hydrogel

Fifteen 8-week-old C57BL/6 mice were randomly divided into five groups ($n = 3$). The gait analysis for the mice after intra-articular injection with hydrogels was investigated by means of an automated Animal Gait analysis system (BT60101, Shenzhen Zhongshi Scientific Instrument, China). After anesthesia, the joint cavities of the mice were injected with different kinds of hydrogels. After woke up, the footprints for the spontaneous walking of mice were collected. Prior to the gait recording, the mice were placed freely in a glass platform to walk for 10 min twice per day, which lasted for 5 days to achieve a brief training session. Then, each mouse was placed individually in the walkway and walked freely from one side to the other side while the gait changes were recorded and analyzed by the software. The average area (average area of a paw contacting the glass) and average intensity (average pressure of a paw contacting the glass) were recorded. The left and right hind limbs of experimental mice were examined for the ROM. For the ROM detection, the mice were anesthetized and then the ROM of mice in the left and right hind limbs were checked passively and measured.

## In vivo retention of hydrogel in arthrography

Nine 8-week-old C57BL/6 mice were randomly divided into three groups ($n = 3$), and then the mice were anesthetized with isoflurane. After that, IR780 (Merck, Germany) labeled PTH (1–34) PBS solution, GGA pre-hydrogel solution, and IR780 labeled Lipo@PTH (1–34) in GGA pre-hydrogel solution to make IR780 labeled GLP pre-hydrogel solution were prepared and each kind solution was intra-articularly injected in a mice. At the set time point, the mice were imaged by means of an IVIS at the excitation wavelength of 657 nm and emission wavelength of 747 nm to determine the retention of injected materials.

## ATDC5 cells culture study

**ATDC5 cells culture and behavior with hydrogels.** The ATDC5 chondrogenic cell lines were purchased from KeyGEN BioTECH (Cat#KG445, KeyGEN, China) and cultured in DMEM/F12 (Gibico, US) with 5% FBS (Gibico, US), 1% penicillin and streptomycin (Gibico, US) in a humidified incubator (5% $CO_2$, 37 °C). In addition, the ATDC5 cells are stimulated with insulin (10 µg/ml) according to the instructions for further study. The cells were digested with trypsin, and after resuspension in a cell culture medium with 500 µL cell suspension in a $2 \times 10^4$ cells/well, were seeded in a 24-well plate. The disk hydrogel samples (8 mm in diameter and 1.5 mm in height) were lyophilized, and then the lyophilized samples were sterilized under UV irradiation for 2 h; the sterilized lyophilized hydrogels were rehydrated with culture medium, and then placed in a transwell (one sample per well). The transwell with hydrogel was placed onto each well in the 24-well plate cultured with ATDC5 cells, and finally, 1 mL of cell culture medium was added in each well. After 48 h culture, the medium was removed from each well, and then 200 µL of live/dead detection reagent (Live/Dead cytotoxicity Kit, Cat#L-3224, Invitrogen, US) was added to each well at RT. After incubating in a dark room for 20 min, the reagent was removed, and the cells were washed with PBS three times. The live/dead fluorescent images were taken with a fluorescence microscope system (Leica, Germany). The live/dead cells were counted with ImageJ software (version 1.53a), and statistical analysis was performed.

For the evaluation of the ATDC5 cells proliferation, the hydrogel samples were cultured with cells by means of the same cell culture method as the Live/Dead assessment, and the Cell Proliferation BeyoClickTM EdU-555 (EdU) Kit (Cat#C0075L, Beyotime Biotechnology, China) was utilized to evaluate the cell proliferation. After 48 h of culture, the medium was removed from each well, followed by fixed with 10% neutral formalin. Then, 200 µL of EdU solution was added to each well and incubated in a dark room for 1.5 h at RT, followed by staining with DAPI (Invitrogen, US) following the kit instructions. After being washed with PBS three times, the cells were randomly observed with a fluorescence microscope system. The EdU and DAPI labeled cells were counted with ImageJ software, and the proliferation rate was calculated with red-stained cells number/blue-stained cells ×100%.

The behavior of the ATDC5 cells, including proliferation and apoptosis was evaluated by means of immunofluorescence analysis after being cultured with the GGA and GLP hydrogels using the same cell culture method. After 48 h of culture, the medium was removed from each well, and cells were washed with warmed PBS three times. Then, the cells were incubated with primary antibodies Ki-67 (Cat#AF1738, 1:200, Beyotime Biotechnology, China), c-Fos (Cat#ab222699, 1:1000, Abcam, UK), and PTH1R (Cat#BS2710, 1:200, Bioword Technology, US) at 4 °C for 10 h, and after washing with PBS three times, the cells were incubated with secondary antibody Goat Anti-Rabbit IgG H&L (Cat#ab150077, 1:1000, Abcam, UK) at RT for 2 h. After being washed with PBS three times, the cells were stained with Prolong Gold anti-fade mounting media containing DAPI, followed by sealing with resin. The stained cells were observed and imaged with the confocal laser-scanning microscope at wavelengths of 488 nm (green) and 594 nm (red) (Leica, Germany). The images for the stained proteins and DAPI were taken with the same parameters and the immunofluorescence intensity from fluorescent images was measured by means of ImageJ.

For FCM tests, the ATDC5 cells were cultured with the GGA hydrogel and GLP hydrogel for 48 h by means of the same cell culture method, and then the cells were digested with trypsin. The cells were stained with the Cell Cycle Staining Kit (Cat#CCS012, Multi Sciences, China) following the instructions, and then the cells were measured with FCM and analyzed by FlowJo software (version 10.4) according to the protocol.

For Western Blot tests, the ATDC5 cells were cultured with the GGA hydrogel and GLP hydrogel for 48 h by means of the same cell culture medium. Glyceraldehyde-3-phosphate dehydrogenase (Gapdh) was used to normalize the releative protein expression. The total protein was harvested in radioimmunoprecipitation assay (RIPA) buffer (Cat#P0013B, Beyotime Biotechnology, China), supplemented with the protease inhibitor phenylmethanesulfonyl fluoride (PMSF) (Cat#P6730, Solarbio, China) and phosphatase inhibitors (Cat#P1260, Solarbio, China). The protein concentration was detected using the Bicinchoninic Acid (BCA) Protein Assay Kit (Beyotime, China). The obtained proteins were separated by means of sodium dodecyl sulfate-polyacrylamide gel electrophoresis (SDS-PAGE) and then transferred onto polyvinylidene fluoride (PVDF) membranes using a trans-buffer at 120 V for 1 h. The membranes were incubated overnight at 4 °C with the primary antibodies including PTH1R (Cat#BS2710, 1:1000, Bioword Technology, US), SOX9 (Cat#ab185966, 1:5000, Abcam, UK), MMP13 (Cat# ab84594, 1:1000, Abcam, UK), ADAMTS5 (Cat#ab41037, 1:250, Abcam, UK), and Gapdh (Cat# AC002, 1:5000, ABclonal, China). Then washed with Tris Buffered Saline with Tween-20 (TBST) (Cat#ST673, Beyotime Biotechnology, China) for three times, followed by incubation with HRP Goat Anti-Mouse IgG (H + L) antibody (Cat#AS003, 1:5000, ABclonal, China) for 50 min at RT. After washing TBST for 30 min, the membranes with the proteins were visualized by means of the BeyoECL plus kit (Cat#P0018M, Beyotime Biotechnology, China).

**Behavior of IL-1β-induced ATDC5 cells with hydrogels.** The $1 \times 10^5$ ATDC5 cells were seeded in each well in a 24-well plate. After 24 h of culture, 10 ng/ml of IL-1β ((Cat#211-11B, PeproTechInc, US) was added to the cell culture medium, and the IL-1β-induced ATDC5 cells were obtained after 24 h of culture. Then, the IL-1β-induced ATDC5 cells were respectively cultured with GGA and GLP hydrogels by means of the same cell culture method, and the medium was changed to a cell culture medium (without IL-1β). After 48 h culture, the cells were incubated with primary antibodies including SOX9 (Cat#ab185966, 1:200, Abcam, UK), Bcl-2 (Cat#AF6285, 1:100, Beyotime Biotechnology, China), and BAX (Cat#AF0057, 1:200, Beyotime Biotechnology, China) in a refrigerator at 4 °C for 10 h, and after being washed with PBS three times, the cells were incubated with secondary antibody Goat Anti-Rabbit IgG H&L (Cat#ab150077, 1:1000, Abcam, UK) at RT for 2 h, followed by staining with DAPI and sealing.

The same cell treatment process and culture method were utilized for further FCM assessment, RT-qPCR, and Western Blot testing. The Annexin V-FITC/PI apoptosis kit (Cat#AT101C, Multi Sciences, China) was applied according to the manual instructions.

For the RT-qPCR assay, primers were designed by means of the Primer bank and double chain were checked by means of BLAST from the NCBI (Supplementary Table 2). After co-culture with hydrogel samples, the cells were digested with trypsin and the mRNA was extracted with Trizol (Invitrogen, US) and RT-qPCR used TaqMan reagents (Takara, Japan). The reactions were performed on the Light Cycler 480 system (Roche Diagnostics, Mannheim, Germany). The expression levels of genes (Sox9, Col2a1, Acan, Adamts5, COX2, and iNOS) were detected and Gapdh was used to normalize the releative gene expression. The final data of the experiment groups were normalized with the control group.

The IL-1β-induced ATDC5 cells in each well were respectively incubated with inhibitors, H89 (Cat#HY-15979A, MCE, US), AG490 (Cat#HY-12000, MCE, US), LY294002 (Cat#HY-10108, MCE, US), and after 6 h culture the GGA and GLP hydrogels were respectively cultured with these cells. After 24 h culture, the samples and medium were removed. Then, the cells were incubated with primary antibodies including p-PI3K (Cat#17366, 1:1000, CST, US), PI3K (Cat#ab191606, 1:1000, abcam, UK), p-AKT (Cat#4060, 1:1000, CST, US), AKT

(Cat#4691, 1:1000, CST, US), ADAMTS5 (Cat#ab41037, 1:250, abcam, UK), and Gapdh (Cat#AC002, 1:5000, ABclonal, China) to evaluate the proteins expression.

**Enzyme-like immunosorbent assay analysis.** For the ELISA test, the IL-1β-induced ATDC5 cells were cultured with hydrogel samples by means of the same cell culture method. After 48 h of culture, the cell culture medium supernatant was obtained by means of high-speed centrifugation for the ELISA test. The inflammatory mediators, IL-6 and TNF-α were evaluated by means of the corresponding detecting kits following the kit instruction; IL-6 kit (Cat#EK206-3-AW1, Multi Sciences, China) and TNF-α kit (Cat#EK282HS-3-AW1, Multi Sciences, China) were used for ELISA tests, and the final presented data for the experiment groups were normalized with the control group.

**ROS-scavenging property of hydrogels cultured with IL-1β-induced ATDC5 cells.** The chondrocytes were treated with IL-1β, and then co-cultured with different samples. The co-culture method was the same as the aforementioned. After the samples were co-cultured with IL-1β-induced chondrocytes for 48 h, the medium was removed from each well, and then the cells were washed with warmed PBS three times. Subsequently, the chondrocytes were stained with a 2′,7′-dichlorodihydrofluorescein diacetate (DCFH-DA) detecting kit (Cat#S0033M, Beyotime Biotechnology, China) following the instructions to detect the ROS level of cells. The stained cells were observed by means of a fluorescence microscope.

For all of the aforementioned cell culture studies, three times of independent replicate experiments were performed.

### In vivo animal study

**Mice and surgical procedures.** Forty 8-week-old C57BL/6J mice were randomly divided into five groups. The mice were anesthetized with isoflurane and then the DMM surgery was performed[42]. After DMM surgery for one week, same volume of each sample was intra-articularly injected into the joint cavities. Notably, the GLP pre-hydrogel was intra-articularly injected and the GLP hydrogel was formed in situ. After, two times of intra-articular injection of the samples were performed at week 4 and week 7. At a set time point (week 10), the mice were sacrificed with an overdose of $CO_2$, and knees were obtained for further evaluation.

**Macroscopical observation and Micro-CT detection.** After implantation for the set time periods, the mice were euthanized, and the knee joints were harvested for macroscopical observation and micro-CT detection. A zoom-stereoscopic microscope (Zeiss, Germany) was used for macroscopic observation of knee joints. Then, the knee joints were scanned by means of a micro-CT (viva CT 80, SCANCO MEDICAL, Switzerland) with a voxel size of 10.4 μm, energy/intensity of 70 kV, 114 μA, and 8 W. The ROI for bone parameter analysis was defined between the SCB plate and epiphysis, and the bone parameters of the SCB including bone mineral density (BMD, mg/cm$^3$), trabecular number (Tb.N), and structure model index (SMI) were calculated.

**Histological assessment.** The obtained knees were fixed in 10% neutral formalin for 48 h and then the knees were decalcified in 10% ethylene diamine tetraacetic acid (EDTA) (pH 7.4) in a shaker (37 °C, 80 rpm) for 5 days. After gradient dehydration, the knees were embedded with paraffin and were cut along the coronary plane at 4 μm for further histological staining, including Hematoxylin and Eosin (H&E) stain kit (Cat#G1120, Solaibao, China), Toluidine Blue-O stain kit (Cat#G2543, Solaibao, China), Safranin-O and fast green stain kit (Cat#G1371, Solaibao, China), Masson's stain kit (Cat#G1340, Solaibao, China), and Alicen blue stain kit (Cat#G1560, Solaibao, China).

The scores to indicate the cartilage wear degree were evaluated by means of the two blind independent investigators.

**Immunohistochemical staining.** The immunohistochemical stainings including PTH1R (Cat#BS2710, 1:400, Bioword Technology, US), SOX9 (Cat#ab185966, 1:1000, abcam, UK), ACAN (Cat#13880-1-AP, 1:500, Proteintech, China), COLIIA1 (Cat#28459-1-AP, 1:500, Proteintech, China), MMP13 (Cat#18165-1-AP, 1:500, Proteintech, China); ADAMTS5 (Cat#bs-3573R, 1:200, Bioss, China), Bcl-2 (Cat#68103-1-Ig, 1:500, Proteintech, China), and BAX (Cat#AF0057, 1:200, Beyotime Biotechnology, China). were performed with 4 μm in thick sections. The stained slices were imaged with a microscope, and the positive areas in the representative images were statistically summarized by ImageJ. Meanwhile, the organs, including the heart, liver, spleen, pulmonary, and kidney of the experimental animals were obtained for H&E staining to evaluate the biosafety.

**Statistics analysis.** In vitro cell culture experiments were performed independently in triplicate and the data were represented as mean ± standard deviation (SD). The GraphPad Prism (Version 9.0.2) software was utilized to present the statistical analysis data. Two-group comparisons were evaluated by unpaired two-tailed Student's $t$ tests, and multiple comparisons were performed by one-way analysis of variance (ANOVA). A $p$-value less than 0.05 was regarded as a statistically significant difference.

### Reporting summary

Further information on research design is available in the Nature Portfolio Reporting Summary linked to this article.

## Data availability

All the data of this study are available within the article and Supplementary Information files or from the corresponding author on request. Source data are also available at https://doi.org/10.6084/m9.figshare.22634641. Source data are provided with this paper.

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

## Acknowledgements

This study was supported by grants from National Natural Science Foundation of China (H.Z.: 82172432; Y.Q.C.: 82202664), Guangdong Basic and Applied Basic Research Foundation (H.Z.: 2022B1515120046; P.L.: 2021A1515220037; P.L.: 2022A1515220165), Shenzhen Key Laboratory of Orthopaedic Diseases and Biomaterials Research (H.Z., Y.Q.C., P.L., J.W., and F.Y.: ZDSYS20220606100602005), "San-Ming" Project of Medicine in Shenzhen (H.Z., and D.W.: SZSM201612092) and Shenzhen Science and Technology Program (H.Z.: JCYJ20220818102815033, KCXFZ20201221173411031, JCYJ20210324110214040 and P.L.: JCYJ20210324105806016). We would also like to thank the Department of Bone and Joint Surgery, Department of Pathology, and Central Laboratory of Peking University Shenzhen Hospital.

## Author contributions

G.Q.L., P.L., Y.Q.C., and H.Z. conceptualized the study. G.Q.L., S.L., Y.X.C., and J.Z., conducted investigation and curated data. G.Q.L., S.L., and Y.X.C. wrote the original draft. H.H.X., J.W., F.Y., A.X., and D.W. visualized data. A.U., P.L., Y.Q.C., and H.Z. were involved in reviewing & editing. P.L., Y.Q.C. and H.Z. supervised research. H.Z. administrated the project. P.L., Y.Q.C., and H.Z. acquired funding.

## Competing interests

The authors declare no competing interests.
