## [Peer Review File · Nature Communications]

REVIEWER COMMENTS

Reviewer #1 (Remarks to the Author):

Overall, the researchers have presented an interesting approach to the treatment of OA, via hydrogel based, liposome-loaded small peptide as an injectable biomaterial.

The in vitro and in vivo results have presented positive results, supporting the potential application of this interesting biomaterial.

That said, the current form of the manuscript mainly focused on the presentation of the results demonstrating the effectiveness of the hydrogel treatment, while some of the physicochemical characterization of the hydrogel system may need more data to support.

For instance, the loading of the peptide into the liposomes was not very clear in terms of the optimization of the conditions? As well as the loading of liposomes into the hydrogel system. It may be helpful if the researchers could provide more of these characterizations as appendix/supp info. This info would greatly facilitate others to reproduce the current work by the researchers.

One technical query - can the authors explain why the liposome size measurement was reported as Intensity in the figure?

Reviewer #2 (Remarks to the Author):

In this manuscript, the authors utilize a novel gallic acid-grafted gelatin injectable hydrogel loaded with liposomes-anchored PTH to construct the sustained release system of drugs composed of liposomes and hydrogels which use enzyme-linking method to control PTH release, aimed to alleviate osteoarthritis progress and protect chondrocytes degradation. Here we appreciate the authors for their efforts, however there are some questions that we find in reading the manuscript, which is listed as follow:

Major:

1. There have been many reports on the sustained release system of drugs composed of liposomes and hydrogels. The authors need to elaborate further on the novelty of this manuscript. "Injectable hydrogel loaded with liposomes-anchored teriparatide (GGA@Lipo@PTH: GLP) was developed with an enzyme-linking method to control PTH release, " and so on. Can you elaborate on your innovation?

2. The hydrogel used by the authors is injected into the joint cavity to form a gel. Does the solid hydrogel filling the joint cavity affect joint motion? If the knee motion of mice is normal, is it because the hydrogel failed to form the gel? The authors need to verify two important questions: 1. Demonstrate successful colloidal formation of hydrogels in the joint through anatomy. 2. The effect of hydrogels with different crosslinking degrees on the knee motion needs to be verified.

3. In Fig 3-5, The authors need to add a blank control group of hydrogel to make the conclusions more rigorous.

Minor :

1. In the Abstract part. The acronym " GGA@Lipo@PTH: GLP " needs to be clarified more clearly. We need to read the introduction of this manuscript to understand what this abbreviation means.

2. In the Introduction part. The authors need to introduce in more detail the research progress of GGA hydrogel and the reason why the author chooses this hydrogel.

3. In Page 4, Line 123, "especially in the ... consume the ROS", Why has the GGA group not shown any therapeutic effect in animal studies?

4. Please provide the Western data with whole membrane and shown with marker ladder.
5. In Fig 5(d1), Why did the control group have such a high rate of apoptosis? Are there any problems with the consumables or environment used to nourish the cells?
6. Why did the authors start injecting hydrogel at 1 week after modeling? Rather than starting treatment after mice have significant OA symptoms? After all, this is more consistent with the clinical treatment scenario.

Reviewer #3 (Remarks to the Author):

The technology reported here is interesting, although the novelty is difficult to evaluate. The use of both an in vitro model (ATDC5 chondrocyte-like cell line) and an in vivo model (DMM) of post-traumatic osteoarthritis (OA) seems appropriate for initial pre-clinical testing. However, there are many errors of English expression (almost every sentence) and the entire manuscript needs to be rewritten by a first-language English speaker, who also understands the science. In addition to the Specific comments listed below, there are several conceptual issues that the authors need to consider if they wish to get this work published:

1. Teriparatide, which is used clinically to treat osteoporosis, has already been tested in pre-clinical models for treatment of post-traumatic OA. For example, Sampson et al., Teriparatide as a chondroregenerative therapy for injury-induced osteoarthritis. *Sci Transl Med.* 2011 Sep 21;3(101):101ra93. (Note that this is the full study reported in abstract form in ref 16 cited in this manuscript.) That study showed not only chondroregenerative effects but also inhibition of cartilage degradation. Importantly, they showed effects on bone.
2. It is not possible to consider effects of teriparatide on cartilage without considering effects on bone because of cross-talk.
3. Part of the authors' rationale seems to be that teriparatide could have anti-inflammatory effects, but inflammatory mediators were not measured in these models.
4. The examination of a full range of anabolic and catabolic markers needs to be done in these models to fully profile the regenerative and chondroprotective effects on chondrocytes in vitro and after intraarticular treatment.
5. Justification of using ATDC5 cells is unclear. This is a mouse cell line that can undergo hypertrophy under certain culture conditions and have been used to examine hypertrophy markers responsive to PTH. Why not used primary human chondrocytes to make the study more clinically relevant.
6. The rationale for examining PI3K/AKT signaling is not clear and it is not well linked to a mechanism controlling regenerative or chondroprotective effects of teriparatide.
7. There are several reports from different laboratories showing the effects of intraarticular gels on OA and inflammatory arthritis models. The manuscript would benefit from an extensive literature search.

Specific comments:

1. Abstract:

- a. Teriparatide and PTH are not the same but seem to be used interchangeably throughout the manuscript. Please define each here and at first use in Introduction. Here is the definition obtained from Google: "Teriparatide is a form of parathyroid hormone (PTH) consisting of the first (N-terminus) 34 amino acids, which is the bioactive portion of the hormone."
- b. Lines 44-46: This sentence needs rewriting: "Further, intraarticular injection of GLP in mice subjected to destabilization of medial meniscus (DMM), a model of OA, confirmed the controlled release of the active form of PTH, the retention of glycosaminoglycans in the cartilage matrix, and the protection of the cartilage from degradation. (Note that "secretion of GAGs may be interpreted as loss of proteoglycans from the cartilage matrix.)"
- c. Please be more specific about which proteins were stimulated or inhibited by the GLP hydrogel and how the PI3K/AKT signaling was specifically involved in regulating each protein.

2. Introduction: This section has many problems with English expression that interfere with understanding the science. Only a few are mentioned below.

- Line 77: OA is not "actually a pro-inflammatory situation. Early events are induced by adverse mechanical loading where chondrocytes can respond by producing catabolic and inflammatory mediators. Inflammation due to synovitis, when present, can promote progression.
- Line 78-79: What does this mean: "irritating the overexpression of transcriptase"?
- Lines 82-83: What comes first? Frictional wear-and-tear of cartilage is viewed to promote the production of proteinases that degrade collagen and proteoglycans.
- Lines 88-89: "promoting secretion of GAGs" is not precise as this can mean loss of GAGs from cartilage. Do you mean that "promotion of synthesis of GAGs" would be desirable. This implies that there is promotion of regeneration or repair. Is this what you mean to say?
- Lines 103-104: It is not true that "very few studies" have been done to maintain normal function and promote repair (NOT "secretion of GAGs).
- Line 114: This sentence does not make any sense: "...the mechanism of PTH to the positive of effect to chondrocyte...". In addition, that this is "still uncertain" is not correct because how PTH acts during chondrogenesis and terminal differentiation during skeletal development is well worked out.
- It is not clear that "NP" or "NPH" is a standard abbreviation for natural polysaccharides. Please spell out in all instances.

3. Results: There are instances of poor grammar, missing verbs or incorrect agreement, convoluted word order, overuse of abbreviations, etc. that make it difficult to evaluate the results described. I gave up trying to write them all here.

- Figure 1: it is unclear whether "GAG" and "ECM" on the scheme were measured as synthetic or degradative products.
- Results pertaining to Figure 2: Since teriparatide used to create the GLP hydrogel is not the full-length PTH, please use a more appropriate abbreviation.
- Line 188 and Figure 3: What is "radiant remained ratio"? Both "radian" and "radiant" are used in the figure legend. Which is correct?
- Lines 209-212, Figure 4g1: The term "bandwidth" is not an accurate way to describe data in Western blots (please do not abbreviate as WB). Does the graph in 4g2 show scans of the Western blots? Do the graphs show scans of replicate (n=4) Western blots from different cultures or 4 scans of the same Western blot?
- Do Fig. 4c2, d2, and e2 show mRNA levels (gene expression)? If not, "expression" should not be used, as it appears that the graphs represent scans of the immunofluorescent images shown above. Both the text and legend are very confusing to the reader. The purpose of measuring all of these proteins needs to be clarified.
- Section 2.5: It is not true that chondrocytes in OA patients "are in the inflammation state", as discussed above. IL-1 is used in in vitro studies to turn on catabolic and inflammatory gene expression so that they can be measured in isolated cells as a model that mimics the destructive environment of progressive OA.
- Lines 231-233, Figure 5: The results are described incorrectly. For all genes, the GLP hydrogel appears to reverse the effects of IL-1. For the anabolic genes, Sox9, Col2a1, and Agc (check correct gene name), IL-1 decreases the mRNA levels and GLP returns them to control levels. In contrast, IL-1 increases Adamts5, a catabolic gene, and GLP blocks this increase. Please check that all results are accurately described.
- The second paragraph of section 2.5 describes the results in Fig. 6, which is accurately indicated in line 234, but the rest of the paragraph incorrectly refers to Fig. 5.
- Results pertaining to Figure 7: Please do not use these non-standard abbreviations (line 265) for the histochemical stains shown in Fig. 7. The results are not well described, as a lot of attention is given to qualitative differences shown in the representative sections, whereas the quantitative data in 7b and 7c are barely explained. What is the "GAG-assumed content"? Please correct "shame" to "sham" control. Basically, these staining patterns show loss of cartilage proteoglycans, including GAGs, due to surgical induction of post-traumatic OA in the DMM model. The GLP injections starting 1 week after surgery attenuated the cartilage damage and prevented the increase in severity of damage measured

as the modified Mankin score.

j. Results pertaining to Figure 8: These data are also not well described and mistakes are similar to those indicated for Figure 7.

k. Lines 305, 309, and 310: What is KOA? This is not a standard abbreviation and was not defined in this manuscript.

4. Discussion: This section is only slightly better written. Some of the information about how the GLP was made and the experimental set-up would be better placed in relevant sections of the Results. The concluding paragraph needs some attention. Notably, the GLP hydrogel did not upregulate MMP13 and ADAMTS5. There was rather a trend towards down-regulation that did not appear to be statistically significant.

5. The conclusion in the Abstract and Discussion that the GLP injectable hydrogel has potential as an OA therapy is poorly supported by the results as they are reported.

6. Methods: The authors should check that these are written clearly. I did not go through this section thoroughly because of time limitations.

Minor: There are many errors of English grammar and expression that are too numerous to mention. Please find an editor who can thoroughly correct the manuscript. Only a few examples are indicated below:

1. Line 38: Correct "verily" to "verify".

2. Line 39: Correct "nanoengineer" to "nanoengineering".

3. Line 84: Please spell out "intraarticular" here and elsewhere instead of using "IA" as abbreviation.

4. Line 122: Correct "gelatin derivates" to "gelatin derivatives", if this is what you mean.

5. Line 137: Correct "...peak was not observed in gelatin...".

6. Lines 160-161: Correct to: "The macroscopic images of the fabricated hydrogels after they were allowed to swell in distilled water for 5 days are presented in Fig. 2(f).

7. ETC., etc...

Point-by-point response to Reviewers' comments

We sincerely appreciate the reviewer's comments to this article, which helped us a lot to improve our work. We've carefully revised the manuscript according to these comments. All the changes are highlighted in red color in the revised manuscript. A point-by-point response has been prepared to address all the concerns as follows (please note that the comments are presented in bold type and our responses are in plain type).

Response to reviewer's comments

Reviewer #1:

1. Overall, the researchers have presented an interesting approach to the treatment of OA, via hydrogel based, liposome-loaded small peptide as an injectable biomaterial. The *in vitro* and *in vivo* results have presented positive results, supporting the potential application of this interesting biomaterial. That said, the current form of the manuscript mainly focused on the presentation of the results demonstrating the effectiveness of the hydrogel treatment, while some of the physicochemical characterization of the hydrogel system may need more data to support. For instance, the loading of the peptide into the liposomes was not very clear in terms of the optimization of the conditions? As well as the loading of liposomes into the hydrogel system. It may be helpful if the researchers could provide more of these characterizations as appendix/supp info. This info would greatly facilitate others to reproduce the current work by the researchers.

[Answer]: Thanks for the evaluation and queries. Actually, in our pilot study the optimization of loading amount of PTH (1-34) in liposome and the amount of LP nanoparticles into this hydrogel system was performed based on the loading efficiency of PTH (1-34) in liposome, the *in vivo* retention of hydrogel in joint cavity. The synthesis of liposomes loaded with PTH

(1-34) was followed the commonly used method (as the cited references in manuscript), and in the preparation process the amount and ratio of components and process parameters, such as sonication intensity and time would affect the size of liposomes and the PTH (1-34) loading efficiency. The optimization criterion in this study is to obtain a higher PTH (1-34) loading efficiency in a suitable range of particle size (e.g. 100 nm in this case) with a low-cost way. The loading amount of liposomes in hydrogel was dependent on the *in vivo* retention of hydrogel in joint cavity and the effective concentration of PTH (1-34) to treatment of OA. We've added more detail for this query in our revised manuscript.

In the last sentence of the second paragraph (line 458-462), the description “Notably, in a pilot study by the present authors, the loading of PTH (1-34) into liposomes and the amount of LP nanoparticles into the hydrogel system were optimized based on the loading efficacy of PTH (1-34) in liposomes, the *in vivo* retention of hydrogel in joint cavity, and the effective concentration of PTH (1-34) for OA treatment.” was added.

In the first paragraph in the Methods part, the details for the calculation of PTH (1-34) loading efficiency in liposome was added (line 586-591): “For the HPLC test, a concentration gradient curve for PTH (1-34) dissolved in DI water was first established; the PTH (1-34) concentration in the LP dialyzed DI water was detected, and the PTH (1-34) loading efficiency in Lipo@PTH (1-34) was calculated according to the following equation: $(M_t - M_n)/M_t \times 100\%$, where M_t is the total amount of PTH (1-34) added to prepare LP nanoparticles, and M_n is related to the non-loaded PTH (1-34) amount detected in dialyzed DI water.”.

2. One technical query - can the authors explain why the liposome size measurement was reported as Intensity in the figure?

[Answer]: Thanks for the query. The Dynamic Light Scattering (DLS) method is a commonly used method to detect the liposome size, and in this method the scattering of the light is correlated with the diffusion level of the liposomes in suspension (small particles diffused faster than the large particles). Actually, the Intensity was not accurate, and the relative intensity (Intensity (a.u.)) was used in our revised manuscript.

Reviewer #2:

In this manuscript, the authors utilize a novel gallic acid-grafted gelatin injectable hydrogel loaded with liposomes-anchored PTH to construct the sustained release system of drugs composed of liposomes and hydrogels which use enzyme-linking method to control PTH release, aimed to alleviate osteoarthritis progress and protect chondrocytes

degradation. Here we appreciate the authors for their efforts, however, there are some questions that we find in reading the manuscript, which is listed as follow:

1. There have been many reports on the sustained release system of drugs composed of liposomes and hydrogels. The authors need to elaborate further on the novelty of this manuscript. “Injectable hydrogel loaded with liposomes-anchored teriparatide (GGA@Lipo@PTH: GLP) was developed with an enzyme-linking method to control PTH release” and so on. Can you elaborate on your innovation?

[Answer]: Thanks for the critical evaluation and suggestion. The novelty is the most important for a publication. We’ve carefully compare this study with the recent reports with regards of sustained release of drugs in liposomes and hydrogels. The novelty of this study was further elaborated in our revised manuscript. To our best knowledge, especially when compare with the recent reports on the OA treatment the PTH (1-34) was the first time to incorporate into a novel fully-biodegradable, biocompatible natural polysaccharide-based hydrogel system to control release of PTH (1-34), followed by the IA injection aimed to alleviate osteoarthritis progress and protect chondrocytes degradation. When further compare with other hydrogels system, this hydrogel compose non-toxic or even natural components (no non-biodegradable components in it), and the fabrication process was mild and controllable, which showed great potential in clinic use. We’ve emphasized the novelty in the Introduction part in our revised manuscript, and the following description was added (in Line 74-78): “In the majority of existing studies, the carrier for bioactive components in OA treatment has been balanced salt solutions or nanoparticles, which largely failed in lab and clinical evaluations due to the rapid diffusion of the injected carrier and fast release of bioactive components, leading to frequent injection^{8,9,10}.”. The new citations were [9] and [10].

Reference:

[8] Yao H, *et al.* Combination of magnesium ions and vitamin C alleviates synovitis and osteophyte formation in osteoarthritis of mice. *Bioactive materials* **6**, 1341-1352 (2021).

[9] Singh H, *et al.* Relative efficacy of intra-articular injections in the treatment of knee osteoarthritis: a systematic review and network meta-analysis. **50**, 3140-3148 (2022).

[10] Kou L, *et al.* Oponized nanoparticles target and regulate macrophage polarization for osteoarthritis therapy: A trapping strategy. **347**, 237-255 (2022).

In Introduction part, the novelty of this study was further emphasized (in Line 140-147): “To the present knowledge, the present study is the first in which PTH (1-34) was incorporated into such a fully-biodegradable, biocompatible natural polysaccharide-based hydrogel system to

control release, with the aim of alleviating osteoarthritis progress and protecting chondrocytes degradation. Compared with another particle-loaded hydrogel system³⁰, the present hydrogel is non-toxic and the major components are natural components (non-degradable components), and the fabrication process is mild and controllable (no need for harsh conditions), having great potential in clinic usage and industry.”.

Reference:

[30] He Y, *et al.* Chondroitin sulfate microspheres anchored with drug-loaded liposomes play a dual antioxidant role in the treatment of osteoarthritis. **151**, 512-527 (2022).

2. The hydrogel used by the authors is injected into the joint cavity to form a gel. Does the solid hydrogel filling the joint cavity affect joint motion? If the knee motion of mice is normal, is it because the hydrogel failed to form the gel? The authors need to verify two important questions: 1). Demonstrate successful colloidal formation of hydrogels in the joint through anatomy. 2). The effect of hydrogels with different crosslinking degrees on the knee motion needs to be verified.

[Answer]: Thanks for the valuable questions. For the question 1), we’ve added the data to demonstrate the colloidal formation of hydrogels after IA injection, and the result was presented in Fig. S3.

In Result part, the following descriptions were included (Line 220-228):

“In situ colloidal formation of hydrogel via intraarticular injection in the joint cavity

The anatomy method was adopted for investigating the colloidal formation of hydrogels in the joint cavity, and the results are presented in Supplementary Fig. S3. Before intraarticular injection, the pre-hydrogel solution or PBS solution (as control) was mixed with nanocarbon. As shown, an observation can be made that after intraarticular injection, the stained area was aggregated in the joint cavity for the hydrogel group, but diffused for the PBS control group with the expression from the joint cavity. Further, a gel-like substance was present in the joint cavity for the hydrogel group, but no solid state-like substance was observed in the PBS group. Such results demonstrate the colloidal formation of the hydrogels after intraarticular injection.”

In Experiment part, the following experiment details were added (Line 654-662):

“The in situ colloidal formation of hydrogel *in vivo*. Six 8-week-old mice (C57BL/6, purchased from Shanghai Research Center for Biomodel Organisms, Shanghai, China) were randomly divided into two groups (n=3). All the animal procedures were conducted following the instructions approved by the Institutional Animal Care and Use Committee (IACUC) of

Peking University Shenzhen Hospital. After anesthesia with isoflurane, the hairs around the knee were shaved. Then, GLP pre-gel mixed with methylene blue was intraarticular injected. The PBS mixed with methylene blue also was also intraarticular injected, which served as a control. After several minutes, the mice were sacrificed and the anatomy of the joint cavity was identified to observe the colloidal formation of the GLP hydrogel.”.

3. In Fig 3-5, The authors need to add a blank control group of hydrogel (GGA) to make the conclusions more rigorous.

[Answer]: Thanks for the constructive suggestions. The blank control group of hydrogel (GGA) was added in these Figures (Fig. 4-6) in our revised manuscript, and the corresponding descriptions was also included.

4. In the Abstract part. The acronym "GGA@Lipo@PTH: GLP" needs to be clarified more clearly. We need to read the introduction of this manuscript to understand what this abbreviation means.

[Answer]: Thanks for the suggestion. We’ve revised this acronym in the Abstract in our revised manuscript (Line 39-43): “In the present study, a novel liposome-anchored teriparatide (Lipo@PTH (1-34)) incorporated gallic acid-grafted gelatin (GGA) injectable hydrogel (GGA@Lipo@PTH (1-34), shortened to GLP) was developed by means of an enzyme-linking method to control the release of PTH (1-34), with the aim of alleviating osteoarthritis progression and protecting chondrocytes degradation.”.

5. In the Introduction part. The authors need to introduce in more detail the research progress of GGA hydrogel and the reason why the author chooses this hydrogel.

[Answer]: Thanks for the suggestion. We’ve cited more recent reports on the GGA hydrogel for tissue regeneration. The reason for choosing GGA hydrogel in this study was the reactive oxygen species (ROS)-scavenging and potential anti-inflammatory properties, and one of important factor for the OA was the inflammation. We’ve cited relevant articles and added the explains in the revised manuscript (Line 153-164): “In the present hydrogel system, gallic acid, a natural antioxidant small phenolic molecule, was first grafted onto gelatin to obtain the GGA through the crosslinking of amino groups in gelatin and carboxyl groups in gallic acid. As a significant gelatin derivative, GGA has attracted considerable attention for biomedical applications. In addition to the attractive properties of gelatin, GGA possesses potential reactive oxygen species (ROS)-scavenging and anti-inflammatory properties ³¹, since the

phenolic hydroxyl group, especially the non-oxidized ones in GGA, might consume the ROS in the damaged tissue area. As such, the focus of recent studies on GGA hydrogels has been on the ROS-scavenging related tissue regeneration and anti-inflammatory properties, including the promotion of diabetic wound healing³², neural repair in traumatic brain injury (TBI)³³ and other tissue regeneration. OA is a disease that is primarily caused by inflammation, and thus, the use of GGA as a matrix material for drug loading and release is desirable.”. The new cited references were [32] and [33].

Reference:

[31] Thi PL, *et al.* In situ forming and reactive oxygen species-scavenging gelatin hydrogels for enhancing wound healing efficacy. *Acta Biomater* **103**, 142-152 (2020).

[32] Hou Y, *et al.* Co-assembling of natural drug-food homologous molecule into composite hydrogel for accelerating diabetic wound healing. **140**, 213034 (2022).

[33] Zhang D, *et al.* Injectable and reactive oxygen species-scavenging gelatin hydrogel promotes neural repair in experimental traumatic brain injury. **219**, 844-863 (2022).

6. In Page 4, Line 123, “especially in the ... consume the ROS”, Why has the GGA group not shown any therapeutic effect in animal studies?

[Answer]: Thanks for the careful review and the query. We’ve revised the description in the Introduction part and provided some other investigations for animal study data in the revised manuscript. However, according to our experience and reported studies the ROS-scavenging and anti-inflammatory property of sole GGA hydrogel was limited, as during the GGA hydrogel preparation process some of phenolic hydroxyl groups would be spontaneously oxidized when it was exposed to the air, which weaken the ROS-scavenging property. In our revised manuscript, the *in vitro* ROS-scavenging property for GLP hydrogel was conducted, and the result was presented in Fig. S5. Accordingly, the GGA hydrogel also showed the ROS-scavenging property, but the efficacy was limited.

In animal study, the immunohistochemical staining for ADAMTS5 and BAX were performed with the tissue sections and the result was showed in Fig. S10, which can reversely reflect the ROS-scavenging property.

The detailed descriptions for experiment and result were also included in the revised manuscript.

For *in vitro* ROS-scavenging property, the detailed experiment part was included (Line 796-803): **“ROS-scavenging property of hydrogels cultured with IL-1 β -induced-ATDC5 chondrocytes.** The chondrocytes were treated with IL-1 β , and then co-cultured with different

samples. The co-culture method was the same as the aforementioned method. After the samples were co-cultured with IL-1 β induced chondrocytes for 48 hours, the medium was removed from each well, and then the cells were washed with warmed PBS three times. Subsequently, the chondrocytes were stained with a 2',7'-dichlorodihydrofluorescein diacetate (DCFH-DA) detecting kit (Beyotime, China) following the instructions to detect the ROS level of cells. The stained cells were observed by means of a fluorescence microscope (Leica, Germany).”.

In Result part, the following description for Supplementary Fig. S5 was added (Line 349-355): “In addition, a DCFH-DA assay was performed to detect the ROS level, and the positive stained fluorescent images are displayed in Supplementary Fig. S5. A higher number of FITC stained cells indicates a higher ROS level. As shown in Supplementary Fig. S5a, the IL-1 β treated sample showed a higher number of stained cells compared with the control group, but after being co-cultured with GGA and GLP hydrogels, the stained cell number decreased, especially for the GLP treated group. The quantitative statistical analysis results in Supplementary Fig. S5b confirm the fluorescent observation results.”.

For animal study, the detailed experiment part was included (in Line 839-846): “The immunohistochemical stainings for PTH1R (PTH/PTHrP-R (L187) polyclonal antibody, 1:400, Bioword Technology, USA), SOX-9 (Anti-SOX9 antibody, 1:2000, ab185966, abcam, USA), AGC (Rabbit Anti-ACAN Polyclonal Antibody, 1:400,bs-1223R, Bioss), COL IIA1 (Rabbit Anti-Collagen II Polyclonal Antibody, 1:200, bs-0709R, Bioss), MMP13 (Rabbit Anti-MMP13 antibody, 1:200, bs-10581R, Bioss, China), ADAMTS5 (Rabbit Anti-ADAMTS5 antibody, 1:200, bs-3573R, Bioss, China), Bcl-2 (Rabbit Anti-Bax antibody, 1:200, bs-0032R, Bioss, China), and BAX (Rabbit Anti-Bax antibody, 1:200, bs-0127R, Bioss, China) were performed with 4 μ m in thick sections.”.

In Result part, the following description for Supplementary Fig. S9 was added (Line 419-428): “Additionally, several anabolic and catabolic biomarkers, including MMP13, ADAMTS5, Bcl-2, and BAX proteins were also investigated in immunohistochemical staining, and the results are presented in Supplementary Fig. S9, which can reversely reflect the ROS-scavenging property of the present hydrogel system as well as the ability to alleviate the progress of OA. From the immunohistochemical staining images, lesser stained brown dots in the cartilage layer (positive stained area) were presented in the expression of MMP13 and BAX proteins for the GGA and GLP treated groups when compared with the PBS treated DMM group. Further, among hydrogels treated groups, the GLP presented the least stained brown area. The statistical

analysis results in Supplementary Figs. S9b-e confirm the immunohistochemical staining observations.”.

7. Please provide the Western data with whole membrane and shown with marker ladder.

[Answer]: Thanks for the suggestions. We’ve provided the whole membrane with marker ladder in the revised manuscript.

8. In Fig 5(d1), why did the control group have such a high rate of apoptosis? Are there any problems with the consumables or environment used to nourish the cells?

[Answer]: Thanks for the queries. As the reviewer suggested, we’ve added a GGA blank control hydrogel and repeated this experiment and the results were presented in the revised manuscript. In the Results part, the following description was added (in Line 288-293): “The FCM results in Fig. 6a show the percentage of Q2 (indicating the apoptosis) of chondrocytes. After being treated with IL-1 β , the apoptosis rate was increased to 1%, but the parameter decreased to 9.93% after treatment with the GGA hydrogel, and even decreased to 9.05% after co-culture with the GLP hydrogel. The apoptosis rate for the GLP group was close to the control group (7%). The statistical analysis results in Fig. 6b show the same tendency as the FCM results.”.

9. Why did the authors start injecting hydrogel at 1 week after modeling? Rather than starting treatment after mice have significant OA symptoms? After all, this is more consistent with the clinical treatment scenario.

[Answer]: Thanks for the questions. The reasons for injecting hydrogel at 1 week after modeling were described as follows. First, the surgical incision healing and removal of sutures for DMM would be finished up after 1 week. Second, it will take a quite long time to have significant OA symptoms after DMM modeling, which will bring more painfulness to the animal as well as increase the costs for the animal feeding. Third, as OA is an irreversible degenerative disease, it may hard to obtain a remarkable effect to retard the OA progress when it has significant OA symptoms, as the primary design of this hydrogel was to alleviate OA progress and protect chondrocytes degradation. Besides, injecting hydrogel at this time point is also a frequently used time point. We’ve added the following description in the revised manuscript (in Line 529-536): “Besides, the reasons for the *in vivo* intraarticular injection of hydrogel at 1 week after modeling were based on the following considerations. First, the surgical incision healing for DMM would be finished up after 1 week. Second, the it will take

a quite long time to have significant OA symptoms after DDM modeling, which will bring more painfulness to the animals and increase the costs. Last but not the least, as OA is an irreversible disease, it may be hard to obtain a remarkable effect to retard the OA progress when it has significant OA symptoms, since the primary design of this hydrogel was to alleviate OA progress and protect chondrocytes degradation.”.

Reviewer #3:

The technology reported here is interesting, although the novelty is difficult to evaluate. The use of both an *in vitro* model (ATDC5 chondrocyte-like cell line) and an *in vivo* model (DMM) of post-traumatic osteoarthritis (OA) seems appropriate for initial pre-clinical testing. However, there are many errors of English expression (almost every sentence) and the entire manuscript needs to be rewritten by a first-language English speaker, who also understands the science. In addition to the Specific comments listed below, there are several conceptual issues that the authors need to consider if they wish to get this work published.

[Answer]: Thanks for the evaluation and suggestions to this manuscript. We've emphasized the novelty of this study in the Introduction part. In Introduction part, the novelty of this study was further emphasized (Line 140-147): “To the present knowledge, the present study is the first in which PTH (1-34) was incorporated into such a fully-biodegradable, biocompatible natural polysaccharide-based hydrogel system to control release, with the aim of alleviating osteoarthritis progress and protecting chondrocytes degradation. Compared with another particle-loaded hydrogel system³⁰, the present hydrogel is non-toxic and the major components are natural components (non-degradable components), and the fabrication process is mild and controllable (no need for harsh conditions), having great potential in clinic usage and industry.”. The English expression in the entire manuscript was revised by a professional institution.

1. Teriparatide, which is used clinically to treat osteoporosis, has already been tested in pre-clinical models for treatment of post-traumatic OA. For example, Sampson et al., Teriparatide as a chondroregenerative therapy for injury-induced osteoarthritis. Sci Transl Med. 2011 Sep 21;3(101):101ra93. (Note that this is the full study reported in abstract form in ref 16 cited in this manuscript.) That study showed not only chondroregenerative effects but also inhibition of cartilage degradation. Importantly, they showed effects on bone.

[Answer]: Thanks for the constructive and valuable comments. We agree that the progress of OA would promote the cartilage degradation, which would also affect the subchondral bone (SCB), since the cross-talk between cartilage and its SCB. OA is a whole joint disorder including the SCB and synovium. In the OA situation, the chondrocytes would secrete the catabolic and inflammatory mediators, which result in the synovitis, promoting the progress of OA.

The cross talk between cartilage and the SCB is of importance in the OA, and many reports confirmed this phenomenon. In our revised manuscript, the micro-CT data for the tibia plates were included, and the results were presented in Supplementary Fig. S7. The statistical analysis data showed that all the groups displayed no significant difference in these bone parameters.

The detailed experiment part was included (in Line 820-830): “**Macroscopical observation and micro-CT detection.** After implantation for the set time period, the mice were euthanized, and the knee joints were harvested for macroscopical observation and micro-CT detection. A zoom-stereoscopic microscope (Leica, Germany) was used for macroscopic observation of knee joints. Then, the knee joints were scanned by means of a microcomputer tomography (micro-CT, vivo CT 80, SCANCO MEDICAL, Switzerland) with a voxel size of 10.4 μm , energy/intensity of 70 kV, 114 μA , and 8 W. The region of interest (ROI) for bone parameter analysis was defined between the SCB plate and epiphysis, and the bone parameters of the SCB including bone mineral density (BMD, mg/cm^3), trabecular connection density (Conn.D), trabecular number (Tb.N), trabecular thickness (Tb.Th, μm), trabecular separation (Tb.Sp, μm), structure model index (SMI), and degree of anisotropy (DA) were calculated.”.

In Result part, the following description for Supplementary Fig. S7 was added (in Line 365-373): “*Micro-CT assay of the tibial plateau*

The macrographic images and the micro-CT data are presented in Supplementary Fig. S7. According to Supplementary Fig. S7a, the GLP treated surface of tibia plateaus exhibited lesser wear or tear trace compared with the other groups (the red arrow represents the wear and tear areas), while the three-dimension images of micro-CT in Supplementary Fig. S7b also exhibited the same tendency. The region of interest (ROI) area for micro-CT analysis is shown in Supplementary Figs. S7c-e. The bone parameters are provided in Supplementary Figs. S7f-l, and the results show that there were no significant differences was among the samples in all the bone parameters.”.

2. It is not possible to consider effects of teriparatide on cartilage without considering effects on bone because of cross-talk.

[Answer]: Thanks for the comments. The cross talk between cartilage and the SCB is of importance in the OA, and many reports confirmed this phenomenon. In our revised manuscript, the micro-CT data for the tibia plates were included, and the results were presented in Supplementary Fig. S7. The statistical analysis data showed that all the groups displayed no significant difference in these bone parameters.

3. Part of the authors' rationale seems to be that teriparatide could have anti-inflammatory effects, but inflammatory mediators were not measured in these models.

[Answer]: Thanks for the carefully review to this manuscript. We've included some inflammatory genes including iNOS, IL-6, and TNF- α expression in our revised manuscript to illustrate the anti-inflammatory property of GLP hydrogel. The PCR and ELISA assay results for these genes were added in Fig. 6.

The detailed experiment part was included (in Line 779-795): "For the RT-qPCR assay, primers were designed by means of the Primer bank and double chain checked by means of BLAST from the NCBI (Supplementary Table S1). After 48 hours of culture with hydrogel samples, the cells were digested with trypsin, and the mRNA was extracted with Trizol (Life Technologies Co., Carlsbad, USA) and RT-qPCR used TaqMan reagents (Takara, Otsu, Japan). The PCR reactions were performed on the Light Cycler 480 system (Roche Diagnostics, Mannheim, Germany). The expression levels of genes (SOX 9, COL IIA1, AGC, ADAMTS5, COX2, and iNOS) were detected, and the final data of the experiment groups were normalized with the control group.

Enzyme-like immunosorbent assay (ELISA) analysis. For the ELISA test, the IL-1 β -induced-ATDC5 chondrocytes were cultured with hydrogel samples by means of the same cell culture method as aforementioned. After 48 hours of culture, the cell culture medium supernatant was obtained by means of high-speed centrifugation for the ELISA test. The inflammatory mediators, IL-6 and tumor necrosis factor- α (TNF- α) were evaluated by means of the corresponding detecting kits following the kit instruction; IL-6 kit (EK206/3-01, LIANKE BIO, China) and TNF- α kit (EK282/4-01, LIANKE BIO, China) were used for ELISA tests, and the final presented data for the experiment groups were normalized with the control group."

In Result part, the following description for Fig. 6 was added (in Line 302-311): "The RT-qPCR results in Figs. 6j-n display that the GLP hydrogel exhibited higher levels of SOX9, COL IIA1, and AGC gene expression when compared with the GGA hydrogel and IL-1 β treated

control group, and such upregulated genes were the anabolic markers. The expression levels of catabolic genes, ADAMTS5 and iNOS for the GLP hydrogel treated group were lower than those of the GGA hydrogel treated group and the IL-1 β treated control group. The ELISA results for the expression of inflammatory mediators, IL-6 and TNF- α in Figs. 6o and 6p show that the IL-6 and TNF- α expression levels for the GLP hydrogel treated group were lower than those of the GGA hydrogel treated group and the IL-1 β treated control group. Such results indicate the potential anti-inflammatory effect of the GLP hydrogel.”.

4. The examination of a full range of anabolic and catabolic markers needs to be done in these models to fully profile the regenerative and chondroprotective effects on chondrocytes *in vitro* and after intraarticular treatment.

[Answer]: Thanks for the suggestion. Although the pathogenesis of OA is not completely understood, the inflammation would usually lead to the imbalance between anabolic and catabolic processes. We’ve added some anabolic and catabolic markers expression in *in vitro* and *in vivo* studies. In *in vitro* study, as shown in revised Fig. 6 the anabolic markers, including SOX9, COL IIA1, and AGC genes and catabolic markers, such as ADAMTS5 and iNOS were investigated when the IL-1 β induced ADTC5 chondrocytes co-cultured with hydrogels.

In the Results part, the following description for *in vitro* study was added (in Line 302-311): “The RT-qPCR results in Figs. 6j-n display that the GLP hydrogel exhibited higher levels of SOX9, COL IIA1, and AGC gene expression when compared with the GGA hydrogel and IL-1 β treated control group, and such upregulated genes were the anabolic markers. The expression levels of catabolic genes, ADAMTS5 and iNOS for the GLP hydrogel treated group were lower than those of the GGA hydrogel treated group and the IL-1 β treated control group. The ELISA results for the expression of inflammatory mediators, IL-6 and TNF- α in Figs. 6o and 6p show that the IL-6 and TNF- α expression levels for the GLP hydrogel treated group were lower than those of the GGA hydrogel treated group and the IL-1 β treated control group. Such results indicate the potential anti-inflammatory effect of the GLP hydrogel.”, and in Line 332-344: “The expression levels for the anabolic marker protein SOX9 and the catabolic marker protein ADAMTS5 were also investigated. From the SOX9 blot in Fig. 6q, the IL-1 β treated group showed a higher SOX9 protein expression when compared with the control group, but such tendency was reversed when the IL-1 β induced ATDC5 cells were cultured with the GLP hydrogel. In addition, with the addition of the PI3K inhibitor (LY294002) the SOX9 protein expression was further increased. Such findings indicate that the GLP hydrogel would promote the synthesis of the ECM, especially in the IL-1 β -induced chondrocytes. Moreover,

the catabolic marker protein ADAMTS5 blot in Fig. 6q shows that the ADAMTS5 protein expression IL-1 β treated group was higher than that of the control group, but the ADAMTS5 protein expression was decreased after being treatment with the GLP hydrogel. Further, with the addition of the PI3K inhibitor (LY294002), the ADAMTS5 protein expression was further increased when compared with the group treated solely with GLP. Such findings indicate that the GLP hydrogel would inhibit the degradation of the ECM.”.

For *in vivo* study, the immunohistochemical stainings were performed, and the result (in Supplementary Fig. S9) description was added (in Line 419-428): “Additionally, several anabolic and catabolic biomarkers, including MMP13, ADAMTS5, Bcl-2, and BAX proteins were also investigated in immunohistochemical staining, and the results are presented in Supplementary Fig. S9, which can reversely reflect the ROS-scavenging property of the present hydrogel system as well as the ability to alleviate the progress of OA. From the immunohistochemical staining images, lesser stained brown dots in the cartilage layer (positive stained area) were presented in the expression of MMP13 and BAX proteins for the GGA and GLP treated groups when compared with the PBS treated DMM group. Further, among hydrogels treated groups, the GLP presented the least stained brown area. The statistical analysis results in Supplementary Figs. S9b-e confirm the immunohistochemical staining observations.”.

5. Justification of using ATDC5 cells is unclear. This is a mouse cell line that can undergo hypertrophy under certain culture conditions and have been used to examine hypertrophy markers responsive to PTH. Why not used primary human chondrocytes to make the study more clinically relevant.

[Answer]: Thanks for the good suggestion. ATDC5 cell line is derived from mouse teratocarcinoma cells, which is more stable than primary human chondrocytes and has frequently used in chondrogenesis related studies. The more stable cell line is more favorable to study the signal pathway. As far as we know, in recent years over 200 publications utilized this cell line to investigate the relevant studies. In our further study, we will consider to use the primary human chondrocytes to make it more clinically relevant. We’ve added the explanation in the Discussion part in revised manuscript (in Line 540-543): “Second, the ATDC5 cell line was used in the present study as the cell line is more stable than the human chondrocytes, especially for signal pathway research. Human chondrocytes will be considered for future research, so as to be more clinically relevant.”.

6. The rationale for examining PI3K/AKT signaling is not clear and it is not well linked to a mechanism controlling regenerative or chondroprotective effects of teriparatide.

[Answer]: Thanks for your careful review and suggestion. We've added some RT-PCR and Western blots results to further illustrate the potential signaling pathway, and a mechanism scheme was added in revised manuscript.

In Results part, the following description was added (in Line 311-348): “The Western blot results in Figs. 6q and 6r show the expression levels of relevant key proteins in the PI3K/AKT signaling pathway when the IL-1 β treated ATDC5 chondrocytes were cultured with the GLP hydrogel with or without different inhibitors. The potential mechanism is presented in Fig. 7. The expression levels of key proteins, p-P13K, PI3K, p-AKT, and AKT in the PI3K/AKT signaling pathway were investigated. As shown in Fig. 6q, the GLP treated group showed a higher p-P13K protein expression as compared with the IL-1 β treated and control groups, and with the addition of JAK2 inhibitor (AG490) or PI3K inhibitor (LY294002), the p-P13K protein expression was decreased but the parameter did not change with the presence of PKA inhibitor (H89). Such findings verify that the GLP hydrogel would activate the p-P13K protein expression (as shown in Fig. 7). From the PI3K blot, the GLP treated group showed no difference in PI3K protein expression when compared with the IL-1 β treated and control groups, but with the addition of PKA inhibitor (H89) or JAK2 inhibitor (AG490), the PI3K protein expression was increased. At the same time, with the presence of PI3K inhibitor (LY294002), the PI3K protein expression decreased, which indicates that the GLP hydrogel would also activate the P13K protein expression. From the p-AKT blot, the IL-1 β treated and GLP treated groups presented higher levels of p-AKT protein expression compared with the control group, but there were no differences in the protein expression levels between the IL-1 β treated group and the GLP treated group. With the addition of PKA inhibitor (H89) or PI3K inhibitor (LY294002), the p-AKT protein expression increased, but there was no difference versus the sole GLP group with the addition of the JAK2 inhibitor (AG490). Such findings indicate that the GLP hydrogel would activate the p-AKT protein expression. The expression levels for the anabolic marker protein SOX9 and the catabolic marker protein ADAMTS5 were also investigated. From the SOX9 blot in Fig. 6q, the IL-1 β treated group showed a higher SOX9 protein expression when compared with the control group, but such tendency was reversed when the IL-1 β induced ATDC5 cells were cultured with the GLP hydrogel. In addition, with the addition of the PI3K inhibitor (LY294002) the SOX9 protein expression was further increased. Such findings indicate that the GLP hydrogel would promote the synthesis of the

ECM, especially in the IL-1 β -induced chondrocytes. Moreover, the catabolic marker protein ADAMTS5 blot in Fig. 6q shows that the ADAMTS5 protein expression IL-1 β treated group was higher than that of the control group, but the ADAMTS5 protein expression was decreased after being treatment with the GLP hydrogel. Further, with the addition of the PI3K inhibitor (LY294002), the ADAMTS5 protein expression was further increased when compared with the group treated solely with GLP. Such findings indicate that the GLP hydrogel would inhibit the degradation of the ECM. The statistical analyses for such protein expressions in Fig. 6r confirm the blot observation results. The results also verify that the GLP hydrogel is a significant factor in protecting IL-1 β -induced-ATDC5 cells from further OA progression, potentially through the PI3K/AKT signaling pathway.”.

7. There are several reports from different laboratories showing the effects of intraarticular gels on OA and inflammatory arthritis models. The manuscript would benefit from an extensive literature search.

[Answer]: Thanks for the suggestions. We’ve cited several recently reported studies with regard to the hydrogels on OA treatment in the revised manuscript in Discussion part (in Line 455-458): “When compared with recent reported injectable hydrogel systems^{12, 39, 40} for OA treatment, the fabrication process of the present GLP hydrogel was mild, nontoxic, and there is no need for complicated polymerization and other harsh conditions. Thus, the present hydrogel system is desirable for clinical use and industry.”.

New cited reference:

[39] Wei J, *et al.* Hierarchically structured injectable hydrogels with loaded cell spheroids for cartilage repairing and osteoarthritis treatment. *Chemical Engineering Journal* **430**, 132211- (2022)

[40] Tong Z. A., *et al.* An injectable hydrogel dotted with dexamethasone acetate-encapsulated reactive oxygen species-scavenging micelles for combinatorial therapy of osteoarthritis. *Mater. Today Nano.* **17**, 100164 (2021).

8. Abstract 8-1-a. Teriparatide and PTH are not the same but seem to be used interchangeably throughout the manuscript. Please define each here and at first use in Introduction. Here is the definition obtained from Google: “Teriparatide is a form of parathyroid hormone (PTH) consisting of the first (N-terminus) 34 amino acids, which is the bioactive portion of the hormone.”

[Answer]: Thanks for the suggestion. We've corrected all the "PTH" as "PTH (1-34)" in revised manuscript.

8. Abstract 8-1-b. Lines 44-46: This sentence needs rewriting: "Further, intraarticular injection of GLP in mice subjected to destabilization of medial meniscus (DMM), a model of OA, confirmed the controlled release of the active form of PTH, the retention of glycosaminoglycans in the cartilage matrix, and the protection of the cartilage from degradation. (Note that "secretion of GAGs may be interpreted as loss of proteoglycans from the cartilage matrix.)"

[Answer]: Thanks for the suggestion. We've revised this sentence as (in Line 49-52): "Further, intraarticular injection of hydrogels into an OA induced mice model, which was created via destabilization of the medial meniscus (DMM), showed that GLP would promote the synthesis of glycosaminoglycans, and protect the cartilage from degradation." in revised manuscript.

8. Abstract 8-1-c. Please be more specific about which proteins were stimulated or inhibited by the GLP hydrogel and how the PI3K/AKT signaling was specifically involved in regulating each protein.

[Answer]: Thanks for the suggestion. We've revised this sentence as (in Line 46-49): "As revealed through *in vitro* studies, the GLP hydrogel promoted ATDC5 proliferation. In addition, it protected the IL-1 β -induced-ATDC5 from further OA progression by upregulating p-PI3K, p-AKT, and SOX9 while downregulating ADAMTS5 potentially via the PI3K/AKT signaling pathway."

9. Introduction: This section has many problems with English expression that interfere with understanding the science. Only a few are mentioned below. 9-2-a. Line 77: OA is not "actually a pro-inflammatory situation. Early events are induced by adverse mechanical loading where chondrocytes can respond by producing catabolic and inflammatory mediators. Inflammation due to synovitis, when present, can promote progression.

[Answer]: Thanks for the constructive suggestions. We've revised this sentence as follows and added the necessary citations in our revised manuscript (in Line 88-98): "However, as an abnormal situation, OA does not just involve the wear-and-tear of cartilage, and is increasingly regarded as a whole joint disorder involving the remodeling of SCB and synovium¹⁴. The initiation of OA could potentially be induced by adverse mechanical loading, wherein the

ingredients result in a proper response by producing catabolic and inflammatory mediators produced by chondrocytes, which could potentially lead to synovitis or even aggravate the OA progression^{15,16}. The normal environment of chondrocytes would be disrupted in OA with the overexpression of catabolic enzymes, which causes degradation of the extracellular matrix (ECM), such as a decrease in glycosaminoglycans (GAGs)¹⁷ and degradation of normal cartilage¹⁸. The degradation of ECM will increase the frictional wear of the articular cartilage, causing pain and even leading to disability for OA patients.”. The new cited references were [14], [15], [16], and [18].

References:

[14] Ni R., Guo X. E., Yan C. & Wen C. Hemodynamic stress shapes subchondral bone in osteoarthritis: An emerging hypothesis. *J. Orthop. Translat.* **32**, 85-90 (2022). [15] Jiang Y. Osteoarthritis year in review 2021: biology. *Osteoarthritis Cartilage.* **30**, 207-215 (2022).

[16] Zhang H., *et al.* Synovial macrophage M1 polarisation exacerbates experimental osteoarthritis partially through R-spondin-2. *Ann. Rheum. Dis.* **77**, 1524-1534 (2018).

[17] Simental-Mendía M., *et al.* Effect of glucosamine and chondroitin sulfate in symptomatic knee osteoarthritis: a systematic review and meta-analysis of randomized placebo-controlled trials. *Rheumatol. Int.* **38**, 1413-1428 (2018).

[18] Zheng L., Zhang Z., Sheng P. & Mobasheri A. The role of metabolism in chondrocyte dysfunction and the progression of osteoarthritis. *Ageing Res. Rev.* **66**, 101249 (2021).

9. Introduction: 9-2-b. Line 78-79: What does this mean: “irritating the overexpression of transcriptase”?

[Answer]: Thanks for the careful review. We’ve deleted this description in the revised manuscript, and corrected as (in Line 93-96): “The normal environment of chondrocytes would be disrupted in OA with the overexpression of catabolic enzymes, which causes degradation of the extracellular matrix (ECM), such as a decrease in glycosaminoglycans (GAGs)¹⁷ and degradation of normal cartilage¹⁸.”.

9. Introduction: 9-2-c. Lines 82-83: What comes first? Frictional wear-and-tear of cartilage is viewed to promote the production of proteinases that degrade collagen and proteoglycans.

[Answer]: Thanks for the question. We also don’t know which one comes first. As we explained, the OA is not just a single problem of cartilage wear-and-tear, but a whole joint disorder involving SCB and synovium. We’ve revised the description in the Introduction part

(in Line 88-98): “However, as an abnormal situation, OA does not just involve the wear-and-tear of cartilage, and is increasingly regarded as a whole joint disorder involving the remodeling of SCB and synovium ¹⁴. The initiation of OA could potentially be induced by adverse mechanical loading, wherein the ingredients result in a proper response by producing catabolic and inflammatory mediators produced by chondrocytes, which could potentially lead to synovitis or even aggravate the OA progression ^{15,16}. The normal environment of chondrocytes would be disrupted in OA with the overexpression of catabolic enzymes, which causes degradation of the extracellular matrix (ECM), such as a decrease in glycosaminoglycans (GAGs) ¹⁷ and degradation of normal cartilage ¹⁸. The degradation of ECM will increase the frictional wear of the articular cartilage, causing pain and even leading to disability for OA patients.”

9. Introduction: 9-2-d. Lines 88-89: “promoting secretion of GAGs” is not precise as this can mean loss of GAGs from cartilage. Do you mean that “promotion of synthesis of GAGs” would be desirable. This implies that there is promotion of regeneration or repair. Is this what you mean to say?

[Answer]: Thanks for the suggestion. Promotion of synthesis of GAGs would be more accurate. We’ve corrected this sentence as (in Line 101-105): “From such perspective, aside from the aforementioned properties, the ideal injectable biomaterials should possess other attractive properties, such as the ability to promote **synthesis of the extracellular matrix (ECM)**, protect normal function, and promote the proliferation of chondrocytes, which would be an alternative strategy for OA treatment with great potential.”.

9. Introduction: 9-2-e. Lines 103-104: It is not true that “very few studies” have been done to maintain normal function and promote repair (NOT” secretion of GAGs).

[Answer]: Thanks for the suggestion. We’ve corrected this sentence as (in Line 114-123): “However, the focus of recent research has been on the improvement of mechanical properties and chondrogenic-promotive properties of gelation-based injectable hydrogels, namely increasing the crosslinking density by introducing a covalent bond (**usually containing non-biodegradable or even toxic components**) into gelatin hydrogels or loading with drugs (for example, oxidized dextran (ODex)) for cartilage defect repair ²¹. **Despite such research efforts, to the present knowledge, there is a scarcity of studies in which the development of gelatin-based injectable hydrogels (without any toxic and non-degradable components) has been achieved using a mild and simple fabrication method, which can maintain the normal function**

of chondrocytes and promote the synthesis of the extracellular matrix (ECM) in the joint cavity for OA treatment.”.

9. Introduction: 9-2-f. Line 114: This sentence does not make any sense: “...the mechanism of PTH to the positive of effect to chondrocyte...”. In addition, that this is “still uncertain” is not correct because how PTH acts during chondrogenesis and terminal differentiation during skeletal development is well worked out.

[Answer]: Thanks for the comment. We’ve rewrote this sentence as follows (in Line 134-135): “Further, the efficacy of PTH (1-34) in a new delivery system for OA treatment is also worthy of further study.”.

9. Introduction: 9-2-g. It is not clear that “NP” or “NPH” is a standard abbreviation for natural polysaccharides. Please spell out in all instances.

[Answer]: Thanks for the suggestion. We’ve spell out “NPH” in the revised manuscript.

10. Result: 10-3. Results: There are instances of poor grammar, missing verbs or incorrect agreement, convoluted word order, overuse of abbreviations, etc. that make it difficult to evaluate the results described. I gave up trying to write them all here. 10-3-a. Figure 1: it is unclear whether “GAG” and “ECM” on the scheme were measured as synthetic or degradative products.

[Answer]: Thanks for the comments. The language of manuscript have been thoroughly corrected with a professional institution. In the revised Fig. 1, the previous “AGC” was replaced by “Chondrocyte”.

10. Result: 10-3-b. Results pertaining to Figure 2: Since teriparatide used to create the GLP hydrogel is not the full-length PTH, please use a more appropriate abbreviation.

[Answer]: Thanks for the suggestion. The abbreviation for teriparatide have revised as “PTH (1-34)” in the Fig. 2 and total manuscript in our revised version.

10. Result: 10-3-c. Line 188 and Figure 3: What is “radiant remained ratio”? Both “radian” and “radiant” are used in the figure legend. Which is correct?

[Answer]: Thanks for your careful review. The “radiant” is correct, and we’ve corrected it in Figure captions and in text in our revised version.

10. Result: 10-3-d. Lines 209-212, Figure 4g1: The term “bandwidth” is not an accurate way to describe data in Western blots (please do not abbreviate as WB). Does the graph in 4g2 show scans of the Western blots? Do the graphs show scans of replicate (n=4) Western blots from different cultures or 4 scans of the same Western blot?

[Answer]: Thanks for your carefully review and suggestions. The “bandwidth” was changed as “blots”, and the abbreviation “WB” were corrected as “Western blots” as well in our revised manuscript. The Fig. 4 (g2) was the statistical analysis result for protein blots, and the vertical axis for these results was changed as: “Relative protein expression (Fold Control)” in the revised Fig. 5(n) in the revised manuscript.

10. Result: 10-3-e. Do Fig. 4c2, d2, and e2 show mRNA levels (gene expression)? If not, “expression” should not be used, as it appears that the graphs represent scans of the immunofluorescent images shown above. Both the text and legend are very confusing to the reader. The purpose of measuring all of these proteins needs to be clarified.

[Answer]: Thanks for the comments. Yes, they are genes expression of mRNA levels. We’ve corrected as “Relative mRNA expression” in Fig. 6j-n in revised version.

10. Result: 10-3-f. Section 2.5: It is not true that chondrocytes in OA patients “are in the inflammation state”, as discussed above. IL-1 β is used in *in vitro* studies to turn on catabolic and inflammatory gene expression so that they can be measured in isolated cells as a model that mimics the destructive environment of progressive OA.

[Answer]: Thanks for the constructive suggestion. We’ve changed this sentence as (in Line 284-287): “IL-1 β is a commonly used component in *in vitro* studies for stimulating the catabolic and inflammatory gene expression of chondrocytes, and IL-1 β treated chondrocytes can be measured in isolated cells as a cell model to mimic the destructive environment of progressive OA ³⁴.”.

Reference:

[34] Liao C. R., *et al.* Advanced oxidation protein products increase TNF- α and IL-1 β expression in chondrocytes via NADPH oxidase 4 and accelerate cartilage degeneration in osteoarthritis progression. *Redox Biol.* **28**, 101306 (2020).

10. Result: 10-3-g. Lines 231-233, Figure 5: The results are described incorrectly. For all genes, the GLP hydrogel appears to reverse the effects of IL-1. For the anabolic genes, Sox9, Col2a1, and Agc (check correct gene name), IL-1 decreases the mRNA levels and

GLP returns them to control levels. In contrast, IL-1 increases Adamts5, a catabolic gene, and GLP blocks this increase. Please check that all results are accurately described.

[Answer]: Thanks for the suggestions. We've checked and corrected in revised manuscript (in Line 302-311): "The RT-qPCR results in Figs. 6j-n display that the GLP hydrogel exhibited higher levels of SOX9, COL IIA1, and AGC gene expression when compared with the GGA hydrogel and IL-1 β treated control group, and such upregulated genes were the anabolic markers. The expression levels of catabolic genes, ADAMTS5 and iNOS for the GLP hydrogel treated group were lower than those of the GGA hydrogel treated group and the IL-1 β treated control group. The ELISA results for the expression of inflammatory mediators, IL-6 and TNF- α in Figs. 6o and 6p show that the IL-6 and TNF- α expression levels for the GLP hydrogel treated group were lower than those of the GGA hydrogel treated group and the IL-1 β treated control group. Such results indicate the potential anti-inflammatory effect of the GLP hydrogel."

10. Result: 10-3-h. The second paragraph of section 2.5 describes the results in Fig. 6, which is accurately indicated in line 234, but the rest of the paragraph incorrectly refers to Fig. 5.

[Answer]: Thanks for the comments. We're sorry for this carelessness, and the corresponding description was corrected in our revised manuscript (in Line 311-348): "The Western blot results in Figs. 6q and 6r show the expression levels of relevant key proteins in the PI3K/AKT signaling pathway when the IL-1 β treated ATDC5 chondrocytes were cultured with the GLP hydrogel with or without different inhibitors. The potential mechanism is presented in Fig. 7. The expression levels of key proteins, p-P13K, PI3K, p-AKT, and AKT in the PI3K/AKT signaling pathway were investigated. As shown in Fig. 6q, the GLP treated group showed a higher p-P13K protein expression as compared with the IL-1 β treated and control groups, and with the addition of JAK2 inhibitor (AG490) or PI3K inhibitor (LY294002), the p-P13K protein expression was decreased but the parameter did not change with the presence of PKA inhibitor (H89). Such findings verify that the GLP hydrogel would activate the p-P13K protein expression (as shown in Fig. 7). From the PI3K blot, the GLP treated group showed no difference in PI3K protein expression when compared with the IL-1 β treated and control groups, but with the addition of PKA inhibitor (H89) or JAK2 inhibitor (AG490), the PI3K protein expression was increased. At the same time, with the presence of PI3K inhibitor (LY294002), the PI3K protein expression decreased, which indicates that the GLP hydrogel would also activate the P13K protein expression. From the p-AKT blot, the IL-1 β treated and GLP treated

groups presented higher levels of p-AKT protein expression compared with the control group, but there were no differences in the protein expression levels between the IL-1 β treated group and the GLP treated group. With the addition of PKA inhibitor (H89) or PI3K inhibitor (LY294002), the p-AKT protein expression increased, but there was no difference versus the sole GLP group with the addition of the JAK2 inhibitor (AG490). Such findings indicate that the GLP hydrogel would activate the p-AKT protein expression. The expression levels for the anabolic marker protein SOX9 and the catabolic marker protein ADAMTS5 were also investigated. From the SOX9 blot in Fig. 6q, the IL-1 β treated group showed a higher SOX9 protein expression when compared with the control group, but such tendency was reversed when the IL-1 β induced ATDC5 cells were cultured with the GLP hydrogel. In addition, with the addition of the PI3K inhibitor (LY294002) the SOX9 protein expression was further increased. Such findings indicate that the GLP hydrogel would promote the synthesis of the ECM, especially in the IL-1 β -induced chondrocytes. Moreover, the catabolic marker protein ADAMTS5 blot in Fig. 6q shows that the ADAMTS5 protein expression IL-1 β treated group was higher than that of the control group, but the ADAMTS5 protein expression was decreased after being treatment with the GLP hydrogel. Further, with the addition of the PI3K inhibitor (LY294002), the ADAMTS5 protein expression was further increased when compared with the group treated solely with GLP. Such findings indicate that the GLP hydrogel would inhibit the degradation of the ECM. The statistical analyses for such protein expressions in Fig. 6r confirm the blot observation results. The results also verify that the GLP hydrogel is a significant factor in protecting IL-1 β -induced-ATDC5 cells from further OA progression, potentially through the PI3K/AKT signaling pathway.”.

26. Reviewer comment: 10-3-i. Results pertaining to Figure 7: Please do not use these non-standard abbreviations (line 265) for the histochemical stains shown in Fig. 7. The results are not well described, as a lot of attention is given to qualitative differences shown in the representative sections, whereas the quantitative data in 7b and 7c are barely explained. What is the “GAG-assumed content”? Please correct “shame” to “sham” control. Basically, these staining patterns show loss of cartilage proteoglycans, including GAGs, due to surgical induction of post-traumatic OA in the DMM model. The GLP injections starting 1 week after surgery attenuated the cartilage damage and prevented the increase in severity of damage measured as the modified Mankin score.

[Answer]: Thanks for the suggestions. We’ve carefully checked and corrected all these descriptions in revised manuscript (in Line 376-406): “Histological staining assessments for

the knee were performed to investigate the effect of GLP on the cartilage, since the cartilage is sensitive to the progress of OA³⁵. The result was showed in Fig. 8. The results are shown in Fig. 8. The thickness of the cartilage and smoothness of the surface (presented in the magnified images) indicate the degradation degree of the cartilage; a thicker and smoother cartilage indicates lesser degradation of the cartilage in OA. At the same time, the uniformity of the stained cartilage tissue is also one of the essential indicators for evaluating the degradation degree of cartilage in the OA model. More uniformly stained cartilage indicates lesser degradation of the cartilage. In the results, the sham group included normal mice without the DMM model which served as a positive control, and DMM mice injected with PBS served as a negative control. From the histological staining results in Fig. 8a, for H&E staining, the DMM group with PBS presented non-uniform red-stained tissue, and the red-stained tissues were more homogeneous with injection of GGA, LP, and GLP. Among the DMM groups, the GLP treated group presented the most homogeneous red-stained tissue, indicating the least degradation of cartilage. The homogeneity of the stained tissue for the GLP sample was even comparable to the sham group. The TBO and SOFG stainings also exhibited the same tendency as the H&E staining results in terms of the homogeneity of stained tissue. Aside from the homogeneity, the TBO and SOFG stainings can reflect the thickness of stained cartilage. The blue-stained area in TBO staining and the bright red-stained area in SOFG staining represent the stained cartilage, which reflect the thickness of the cartilage. Such results also display the same tendency as H&E staining. Compared with the non-uniform stained area of PBS, GAG, and LP treated groups, the GLP hydrogel treated group presented a more homogenous positive stained cartilage layer. More uniform staining in the cartilage layer indicates a lesser degradation degree of cartilage. The Masson and Alcian Blue staining results in Supplementary Fig. S8 show the same tendency. GAG is the main component of ECM in the cartilage, and can alleviate an OA patient's pain in the knee. According to the present staining results, the GAG-assumed content versus the sham group was calculated and the results are presented in Fig. 8b, and the results confirm the aforementioned knee staining results. In DMM mice, the GLP hydrogel treated group presented the largest GAG assumed content. Moreover, the severity of the joint was evaluated according to the staining results with the principal Modified Mankin score³⁶ shown in Supplementary Table S2, and the results are presented in Fig. 7c. In DMM mice, the GLP hydrogel treated group presented the lowest score.”

27. Reviewer comment: 10-3-j. Results pertaining to Figure 8: These data are also not well described and mistakes are similar to those indicated for Figure 7.

[Answer]: Thanks for the suggestions. We've revised it accordingly in revised manuscript (in Line 409-428): "Immunohistochemical staining for the expression of PTH1R, SOX9, AGC, and COL IIA1 proteins was investigated in the cartilage area after 10 weeks of IA injection, and the results are shown in Fig. 9. From the staining results (Fig. 9a), in DMM mice the GLP treated group and the LP treated hydrogel group exhibited a dark-brown stained color versus light-brown stained color for PBS and GGA treated groups in the PTH1R immunochemical staining images. However, in all other immunochemical staining results, the GLP hydrogel treated group exhibited a dark-brown stained color versus a light-brown stained color for the other samples. The statistical results shown in Figs. 9b-e confirm the results of the immunochemical staining images. The tissue staining results demonstrate that the GLP hydrogel could alleviate the KOA progression by GAG secretion as well as protect against the degradation of cartilage in OA situations. Additionally, several anabolic and catabolic biomarkers, including MMP13, ADAMTS5, Bcl-2, and BAX proteins were also investigated in immunohistochemical staining, and the results are presented in Supplementary Fig. S9, which can reversely reflect the ROS-scavenging property of the present hydrogel system as well as the ability to alleviate the progress of OA. From the immunohistochemical staining images, lesser stained brown dots in the cartilage layer (positive stained area) were presented in the expression of MMP13 and BAX proteins for the GGA and GLP treated groups when compared with the PBS treated DMM group. Further, among hydrogels treated groups, the GLP presented the least stained brown area. The statistical analysis results in Supplementary Figs. S9b-e confirm the immunohistochemical staining observations."

28. Reviewer comment: 10-3-k. Lines 305, 309, and 310: What is KOA? This is not a standard abbreviation and was not defined in this manuscript.

[Answer]: Thanks for the suggestion. The "KOA" has been changed as "OA" in the revised manuscript.

29. Reviewer comment: 11-4. Discussion: This section is only slightly better written. Some of the information about how the GLP was made and the experimental set-up would be better placed in relevant sections of the Results. The concluding paragraph needs some attention. Notably, the GLP hydrogel did not upregulate MMP13 and ADAMTS5. There was rather a trend towards down-regulation that did not appear to be statistically significant.

[Answer]: Thanks for the suggestions. We've deleted the information about how the GLP hydrogel was made in Discussion part. In the revised manuscript, we've added some key genes and proteins expression results including the statistical analysis results in revised Fig. 5 and Fig. 6, which provide more solid data to support the conclusions.

30. Reviewer comment: 12-5. The conclusion in the Abstract and Discussion that the GLP injectable hydrogel has potential as an OA therapy is poorly supported by the results as they are reported.

[Answer]: Thanks for the suggestions. We've included some key genes and proteins expression and corresponding statistical analysis results in the revised Fig. 5 and Fig. 6, which can further support that the GLP injectable hydrogel has the potential ability to OA treatment. In the revised manuscript, the following description was added (in Line 560-570): "The GLP hydrogel would promote the proliferation of ADTC5 chondrocytes by upregulating the expression of key proteins, Ki67, c-Fos, and PTH1R. Also, GLP hydrogel would protect the IL-1 β induced ATDC5 chondrocytes from further progression by upregulating the expression of key anabolic proteins, SOX9, Bcl-2, COL IIA1, AGC, and simultaneously downregulating the expression of key catabolic or inflammatory proteins, BAX, ADAMTS5, iNOS, COX2, IL-6, and TNF- α , which potentially suggested via the PI3K/AKT signaling pathway. The results of *in vivo* intraarticular injection in DMM mice confirm that the GLP hydrogel would protect the cartilage layer from degradation, alleviate OA progression and promote the synthesis of glycosaminoglycan. Finally, the GLP hydrogel would not cause systemic toxicity. Overall, the GLP injectable hydrogel shows great potential in OA treatment."

31. Reviewer comment: 13-6. Methods: The authors should check that these are written clearly. I did not go through this section thoroughly because of time limitations. Minor: There are many errors of English grammar and expression that are too numerous to mention. Please find an editor who can thoroughly correct the manuscript. Only a few examples are indicated below.

[Answer]: Thanks for the suggestion. We've carefully checked again for the Methods, and the language of this manuscript was revised by a professional institution.

32. Reviewer comment: 13-6-1. Line 38: Correct "verily" to "verify".

[Answer]: Thanks for the suggestion. We've corrected all these in the revised manuscript.

33. Reviewer comment: 13-6-2. Line 39: Correct “nanoengineer” to “nanoengineering”.

[Answer]: Thanks for the suggestion. We’ve corrected as “ nanoengineering” in Line 39 in the revised manuscript.

34. Reviewer comment: 13-6-3. Line 84: Please spell out “intraarticular” here and elsewhere instead of using “IA” as abbreviation.

[Answer]: Thanks for the suggestion. We’ve spelled out “intraarticular” in elsewhere in the revised manuscript.

35. Reviewer comment: 13-6-4. Line 122: Correct “gelatin derivates” to “gelatin derivatives”, if this is what you mean.

[Answer]: Thanks for the suggestion. We’ve changed it accordingly.

36. Reviewer comment: 13-6-5. Line 137: Correct “...peak was not observed in gelatin...”.

[Answer]: Thanks for the suggestion. We’ve changed as “In the ¹H NMR spectra, a new characteristic peak was detected at 7.0 PPM which corresponded to the gallic acid in the GGA component, **but such peak was not found in gelatin.** From the ¹H NMR results, the successful grafting of gallic acid on gelatin was verified.” in line 176-178 in the revised manuscript.

37. Reviewer comment: 13-6-6. Lines 160-161: Correct to: “The macroscopic images of the fabricated hydrogels after they were allowed to swell in distilled water for 5 days are presented in Fig. 2(f).

[Answer]: Thanks for the suggestion. We’ve corrected this sentence as (in Line 201-202): “**The macroscopic images of the fabricated hydrogels after being allowed to swell in distilled water for 5 days are presented in Fig. 2f.”.**

38. Reviewer comment: 13-6-7. ETC., etc...

[Answer]: Thanks for the suggestion. We’ve corrected it in the whole manuscript accordingly. We are sincerely grateful for all these valuable and constructive comments or suggestions to this manuscript, which helps us a lot to improve the quality of this manuscript and our further research.

REVIEWER COMMENTS

Reviewer #2 (Remarks to the Author):

We have carefully studied Reviewer #1's questions and the author's answers. We are satisfied with the author's answer to the second question. However, we think the author's answer to the first question is not good enough.

The first question of Reviewer #1 requires the author to provide more physicochemical characterization of the hydrogel system. But the authors did not provide any new experimental data. At the same time, the Reviewer #1 emphasized the need for the author to provide detailed data on liposomes loaded with peptides and hydrogel loaded with liposomes. We believe that Reviewer #1 should be required to provide load rates for peptides and liposomes under different concentration gradients, and drug release curves need to be measured at different concentrations. And based on these experimental data, provide the reason for choosing this final concentration in the manuscript. Because these are important data for this manuscript.

Reviewer #3 (Remarks to the Author):

The authors have responded adequately to my major comments in this revision. The writing is markedly improved, especially in response to specific comments.

1. However, there remain errors that could be caught by copy-editing if available from the journal. If not, the manuscript may have to go back to the authors for minor revision.

2. Also, the use of abbreviations needs to be checked according to journal standards, if available. For example, TBO and SOFG are not defined at first use in the manuscript, but await their definition in the Methods section. In any case, these are non-standard definitions and the names of the stains need to be spelled out. other abbreviations need to be thoroughly checked.

3. The figures and figure legends need some attention to be sure that they are accurately labeled, that abbreviations are defined and that the data findings are summarized. For example, in the stained sections of joints in Figures 8 and 9, the changes due to OA and treatment need to be marked, especially when not obvious.

Point-by-point response to Reviewers' comments

We are appreciative of the insightful comments made on this manuscript by the editor and reviewers. Their comments were instrumental in helping us enhance the quality of our work. We have meticulously revised the manuscript to incorporate these suggestions, and all the revisions are clearly marked in **RED**. To address the concerns raised by the editor and reviewers, we have prepared a detailed point-by-point response. (Please note that the comments are presented in **BOLD** typeface while our responses are in PLAIN text).

Response to reviewer's comments

Reviewer #2:

We have carefully studied Reviewer #1's questions and the author's answers. We are satisfied with the author's answer to the second question. However, we think the author's answer to the first question is not good enough.

The first question of Reviewer #1 requires the author to provide more physicochemical characterization of the hydrogel system. But the authors did not provide any new experimental data. At the same time, the Reviewer #1 emphasized the need for the author to provide detailed data on liposomes loaded with peptides and hydrogel loaded with liposomes. We believe that Reviewer #1 should be required to provide load rates for peptides and liposomes under different concentration gradients, and drug release curves need to be measured at different concentrations. And based on these experimental data, provide the reason for choosing this final concentration in the manuscript. Because these are important data for this manuscript.

[Answer]: Dear Reviewer, thank you so much for your valuable feedback, which is highly appreciated and helps us improve the quality of our research. Our team has carefully addressed your comments in the amended manuscript.

Answer to sub-comment-(1): As per your comment, we have performed additional strong evidence of physicochemical characterization of the developed hydrogels, and the results are shown in Supplementary Fig.S5. In the Results part (line 244-252), the description

“Mechanical properties of hydrogels

The mechanical properties results were presented in Supplementary Fig. S5. As shown in Supplementary Fig. S5b, the rectangular hydrogels would be recovered to their initial state after 360 ° rotation test. From the stretching test results (in Supplementary Fig. S5c, d), these two hydrogels would not be broken until stretching to four times their original length, and from the tensile stress vs. strain curves, the maximum tensile stress for GGA and GLP hydrogels was 20.0 kPa and 17.5 kPa, respectively. The compressive stress vs. strain curves in Supplementary Fig. S5g showed that these two hydrogels would be broken after being compressed down to ca. 75% of their initial height, and the compressive stress for these two hydrogels was ca. 250 kPa.” was added.

In the Methods part (line 722-729), the description **“Mechanical property tests of hydrogels.**

The hydrogels were formed in a man-made mould or syringe to fabricate the rectangular or cylindrical samples for mechanical property tests. For the rotation test, the rectangular samples (100 mm×10 mm×1 mm) were rotated for 360° at RT for 30 seconds. The same size of rectangular samples was used for stretching tests, which were performed with a static and dynamic material testing machine (Moxin electronic tension machine MX-0580, Jiangsu, China) with a stretching speed of 5 mm/min. The cylindrical samples (10 mm in height and 8 mm in diameter) were used for compression testing using the same material testing machine with a compression speed of 5 mm/min.” was added.

Answer to sub-comment-(2): In response to your comment, we have evaluated the PTH (1-34) loading efficiency in liposomes, and the results are displayed in Supplementary Fig. S2. In the Results part (line 184-192), the description “ The loading efficiency of PTH (1-34) in liposomes was characterized with the High Performance Liquid Chromatography-Mass Spectrometry (HPLC-MS, Thermo Explorer 480, US) method. The loading efficiency of PTH (1-34) in liposomes was optimized by inputting different amounts of PTH (1-34) in the same amount of liposome, and the result was presented in Supplementary Fig. S2. When the input

amount of PTH (1-34) was 1.0 mg, the loading efficiency of PTH (1-34) in liposomes was ca. 66.5%, and with the input amount of PTH (1-34) increased to 1.5 mg, the loading efficiency of PTH (1-34) was increased to ca. 72%, but loading efficiency was not further increased when the input amount increased to 2.0 mg. Therefore, the 1.5 mg PTH (1-34) input amount was utilized for the further investigations.” was added.

In the Methods part (line 621-625), the description “Subsequently, 3 mL of DI water dissolved with different amounts, including 1.0, 1.5, or 2.0 mg PTH (1-34) (Cat#HY-P0059, MCE, US) was added to the flask, and then the flask was subjected to sonication at 37 °C for 1 h, forming micron-sized multi-lamellar liposomes. The different input amounts of PTH (1-34) were used to optimize the loading rate of PTH (1-34) in liposomes.” was added.

Answer to sub-comment-(3): The PTH (1-34) release curve for the different concentrations of Lipo@PTH (1-34) nanoparticles incorporated into hydrogels was provided in the revised manuscript, and the results were presented in Supplementary Fig. S3.

In the Results part (line 228-238), the description “ Besides, the different concentrations of Lipo@PTH (1-34) were incorporated into hydrogels to investigate the PTH (1-34) release profile, and the results were presented in Supplementary Fig. S3. As shown in In Vivo Imaging System (IVIS) images of hydrogels (Supplementary Fig. S3a), the higher amount of Lipo@PTH (1-34) incorporated hydrogels (1 wt% and 0.5 wt%) showed higher radiant intensity in comparison with the lower amount of Lipo@PTH (1-34) incorporated hydrogels, and with the increase of immersion time the radiant intensity was gradually decreased. The Lipo@PTH (1-34) (1 wt%) incorporated hydrogel showed the largest radiant intensity in all these detected time points. The statistical analysis result in Supplementary Fig. S3b confirmed the IVIS observation. Meanwhile, the accumulated PTH (1-34) release in immersed phosphate-buffered saline (PBS) solutions (Supplementary Fig. S3c) showed the same tendency as IVIS images in hydrogels.” was added.

In the Discussion part (line 537-544), the description “Note that the IVIS method was used to evaluate the release of PTH (1-34) in this hydrogel system, which would be an alternative method to evaluate the release profile of PTH (1-34). As expected, different amounts of Lipo@PTH (1-34) incorporated into hydrogels presented different PTH (1-34) release concentrations and durations. In the pilot study, the ATDC5 cells cytotoxicity study with the different amounts of Lipo@PTH (1-34) incorporated hydrogels was performed, and the 1

wt% Lipo@PTH (1-34) incorporated hydrogel showed the cytotoxicity to the cells. Therefore, 0.5 wt% Lipo@PTH (1-34) incorporated hydrogel was used for the further study.” was added.

In the Methods part (line 666-687), the description “ The different concentrations of Lipo@PTH (1-34) incorporated into the hydrogel were utilized to assess the PTH (1-34) release curves. For this assessment, 2.0 mg PTH (1-34) and 0.23 mg Fluorescein 5-isothiocyanate (FITC) (Cat# AT101C, Multi Sciences, China) were dissolved in a 2 mL DMSO solution, and after reaction at room temperature (RT) for 24 h, the FITC-labelled PTH (1-34) was synthesized. Then, the reacted solution was dialyzed (the molecular weight of the dialysis bag was 1000 kDa) in DI water for three days to remove the non-reacted FITC and DMOS solvent. During the dialysis process, the PTH (1-34) would not diffuse out of dialysis bag. Finally, a 3 mL FITC-labelled PTH (1-34) DI water solution was obtained. After that, the FITC-labelled PTH (1-34) was utilized for the synthesis of FITC-labelled Lipo@PTH (1-34) nanoparticles following the abovementioned process. Then, different amounts of FITC-labelled Lipo@PTH (1-34) nanoparticles, including 0.125, 0.25, 0.5, and 1 wt%, were respectively added into a 10 wt% GGA solution; after including 5 wt% of TG 200 μ L of each kind of FITC-labelled Lipo@PTH (1-34) mixed pre-hydrogel solutions was added into each well in a 96-well plate. After the formation of the hydrogel, 200 μ L of PBS solution was added into each well and incubated at RT to assess the PTH (1-34) release profile. At the set time points, the PBS in each well was transferred to a new 96-well plate, followed by the IVIS detection of these immersed PBS solutions and hydrogels; after that, 200 μ L of fresh PBS was added to each well until the next set time point. Then, the released PTH (1-34) in PBS solution was calculated from the standard curves of the concentration gradient of FITC-labelled PTH (1-34) in DI water according to the IVIS detection. Finally, the IVIS images for hydrogels and the accumulated PTH (1-34) release profile were obtained.” was added.

Reviewer #3

The authors have responded adequately to my major comments in this revision. The writing is markedly improved, especially in response to specific comments.

1. However, there remain errors that could be caught by copy-editing if available from the journal. If not, the manuscript may have to go back to the authors for minor revision.

[Answer]: Dear Reviewer, many thanks for your positive comment; it means a lot to our team. We appreciate your suggestion regarding the need for copy-editing. We fully agree that an error-free document is essential to ensure the clarity and accuracy of our research. We have taken your suggestions into account and ensured that our manuscript is thoroughly reviewed to address any errors or inconsistencies.

2. Also, the use of abbreviations needs to be checked according to journal standards, if available. For example, TBO and SOFG are not defined at first use in the manuscript, but await their definition in the Methods section. In any case, these are non-standard definitions and the names of the stains need to be spelled out. other abbreviations need to be thoroughly checked.

[Answer]: Dear Reviewer, thank you for taking the time to review our manuscript and for providing valuable feedback. As per your suggestion, we have thoroughly revised the manuscript and spelled out all abbreviations. Also, we have carefully checked the document to ensure that there are no further instances of unexplained or ambiguous abbreviations.

3. The figures and figure legends need some attention to be sure that they are accurately labeled, that abbreviations are defined and that the data findings are summarized. For example, in the stained sections of joints in Figures 8 and 9, the changes due to OA and treatment need to be marked, especially when not obvious.

[Answer]: Dear Reviewer, thank you for your constructive comments on our manuscript. We appreciate your feedback regarding the figures and figure legends and understand the importance of accurately labelling figures and summarizing data findings.

As per your comment, we have revised the figures and marked the degenerated areas of cartilage in Figure 8a and the immunohistochemically positive stained areas in Figure 9a with yellow arrows. In addition, we have marked the degenerated areas in cartilage and the immunohistochemically positive stained areas in Figures S10a, S11a, and S12a in the revised manuscript.

Additionally, we have carefully reviewed all figures in the manuscript to ensure that they are accurately labelled, and any abbreviations used are clearly defined. Also, we ensured that the data findings were adequately summarized in the figure legends.

REVIEWERS' COMMENTS

Reviewer #2 (Remarks to the Author):

Good revision for publication.